# Enhancing Cross-subject Emotion Recognition via Heterogeneous Distribution Augmentation and Collaborative Learning

Wending Xiong [1]   Ruimin Hu [2]   Lingfei Ren [3]   Junhang Wu [4][5]   Mei Wang [1]   Dengshi Li [6]   Mang Ye [1]

## Abstract

Cross-subject emotion recognition aims to improve a model's generalization to previously unseen subjects. Existing methods are mainly built upon domain generalization or data augmentation, but suffer from two major limitations: 1) heavy dependence on modality-specific feature designs–almost exclusively tailored to EEG signals–resulting in limited generalizability; and 2) the widespread assumption of independently and identically distributed data, which restricts the diversity of generated samples. To address these challenges, we systematically analyze the heterogeneous distribution characteristics of emotion data and propose *MixEmo*, a framework that integrates heterogeneous distribution augmentation and collaborative learning. Specifically, a well-trained backbone is used to extract representations and partition them into multiple single-distribution subsets as distribution prototypes. These prototypes are randomly combined to synthesize unseen distributions, thereby enhancing distributional diversity. Finally, heterogeneous distribution collaborative learning jointly optimizes the model across subsets. Extensive experiments demonstrate that *MixEmo* substantially improves generalization performance in cross-subject emotion recognition.

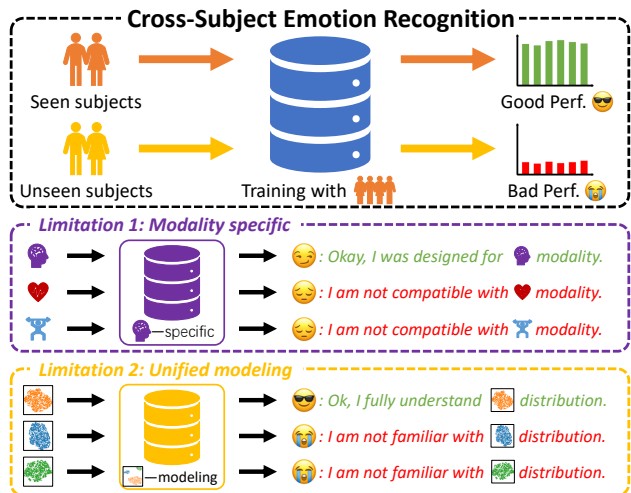

*Figure 1.* Background and Limitations. The goal of cross-subject emotion recognition is to improve a model's generalization to unseen subjects. However, existing approaches suffer from two major limitations: 1) many models rely heavily on modality-specific designs, making them difficult to adapt to other modalities; and 2) most methods assume data are independently and identically distributed (i.i.d.) and perform holistic modeling over heterogeneous datasets, resulting in poor generalization to minority subsets.

## 1. Introduction

Emotion recognition aims to identify an individual's emotional state using various modalities, including EEG signals (Li et al., 2024a; Liu et al., 2024a), text (Shen et al., 2021; Tu et al., 2022), speech (Tan et al., 2022; Wagner et al., 2023), images (Zhu et al., 2023), videos (Du et al., 2021; Zhang et al., 2022b), and societal behaviors(Xiong et al., 2025). Existing models typically demonstrate strong performance on subjects seen during training. However, substantial variability across subjects often leads to a significant degradation in performance when models are evaluated on previously unseen subjects (Shen et al., 2023; Zhao et al., 2021; Cui et al., 2023). This task is referred to as cross-subject emotion recognition (Hong et al., 2025; Ye et al., 2025).

Existing studies typically achieve cross-subject emotion recognition through domain generalization or data augmentation techniques. However, such methods still suffer from two notable limitations, as illustrated in Figure 1. First, most

[1]School of Computer Science, Wuhan University, Wuhan, China [2]School of Cyber Science and Engineering, Wuhan University, Wuhan, China [3]School of Computing and Artificial Intelligence, Southwestern University of Finance and Economics, Chengdu, China [4]School of Computer Science and Technology, Xinjiang University, Urumqi, China [5]Hubei Provincial Key Laboratory of Multimedia and Network Communication Engineering, Wuhan University, Wuhan, China [6]School of Artificial Intelligence, Jianghan University, Wuhan, China. Correspondence to: Ruimin Hu <hrm@whu.edu.cn>, Lingfei Ren <renlf@swufe.edu.cn>.

*Proceedings of the 43rd International Conference on Machine Learning*, Seoul, South Korea. PMLR 306, 2026. Copyright 2026 by the author(s).

existing domain generalization approaches are almost exclusively designed for EEG signals (Ma et al., 2023; Jiménez-Guarneros & Fuentes-Pineda, 2024; Li et al., 2024b; She et al., 2023; Gong et al., 2024). With the rapid advancement of multimodal emotion recognition (Zhang et al., 2022c; Liu et al., 2024b; Cheng et al., 2024), modality-agnostic or weakly modality-dependent models have become increasingly important. In contrast, these carefully handcrafted, modality-specific models are often difficult to transfer effectively across different emotion modalities. Second, although some data augmentation methods achieve a certain degree of modality independence–such as generative augmentation strategies based on Generative Adversarial Networks (GANs) (Su & Lee, 2023; Zhang et al., 2023) and Diffusion Models (DMs) (Zou et al., 2024; Shome et al., 2024)–studies on public emotion datasets indicate that the widely existing class imbalance issue may limit their effectiveness (Wirawan et al., 2022). More critically, data collected in real-world scenarios often fail to satisfy the i.i.d. assumption on which these methods rely. Motivated by the mixed-distribution nature of emotion data, this paper proposes a weakly modality-dependent framework for cross-subject emotion recognition.

To gain deeper insights into the characteristics of emotion datasets, we visualized the distribution of samples in the embedding space (see Section 2.2). The results reveal significant distributional heterogeneity: the samples typically form multiple subsets with markedly different distributions and scales. This suggests that modeling all heterogeneous subsets holistically may limit the model's performance. Therefore, for datasets exhibiting distributional heterogeneity, the heterogeneous subsets should be explicitly modeled and collaboratively learned.

Building on the above analysis, we decompose the research focus of this work into the following two subproblems: I. How can distributional diversity be enhanced under limited data conditions? Data augmentation methods built upon the i.i.d. assumption struggle to handle mixed-distribution scenarios, as they typically generate samples confined to a single or a small number of dominant distributions (Volpi et al., 2018; Thulasidasan et al., 2019). Although instance-level synthesis techniques such as Mixup (Zhang et al., 2017) have demonstrated strong performance in image-related tasks, their direct application to emotion modalities–where samples are strongly constrained–may introduce latent instance conflicts (Zhang et al., 2022a), thereby compromising the validity of the augmented samples. II. How can multiple heterogeneous subsets be jointly learned and integrated into a unified model? Deep learning models generally perform holistic modeling of data according to the downstream task. However, when the training data consist of multiple heterogeneous subsets, such unified modeling tends to be biased toward the dominant distribution (Carlucci et al., 2019;

Grinsztajn et al., 2022), leading to substantial performance degradation on non-dominant subsets.

In this paper, we propose *MixEmo*, a cross-subject emotion recognition framework based on heterogeneous-distribution augmentation and collaborative learning, designed to address the challenges of insufficient distributional diversity and imbalanced learning across heterogeneous subsets. To tackle Problem I, we introduce the **U**nseen **D**istribution **G**eneration (UDG). Leveraging the inherent distributional heterogeneity of emotion data, we first partition the mixed-distribution representation set into multiple single-distribution subsets, which serve as distribution prototypes. Unseen distribution subsets are then generated by randomly combining samples from these prototypes, thereby effectively enhancing the overall diversity of the data. To address Problem II, we design the **H**eterogeneous **D**istribution **C**ollaborative **L**earning (HDCL). For each heterogeneous subset, we randomly split it into a support set and a query set and construct an independent proxy model, while maintaining a shared prediction model. Each proxy model is trained on the support set of its corresponding subset and evaluated on the query set; the resulting query loss is used as a weighting factor in updating the prediction model parameters. Through this collaborative optimization process, the prediction model effectively aggregates knowledge from multiple heterogeneous distributional subsets, ultimately achieving improved generalization. The main contributions of this work are summarized as follows:

- **Comprehensive analysis.** We conduct a thorough review of existing cross-subject emotion recognition methods and perform an in-depth analysis of the distributional heterogeneity exhibited across five emotion datasets. This characteristic exposes the limitations of current data augmentation strategies when applied to mixed-distribution scenarios and provides the foundation for our methodological design.

- **A novel methodology.** Motivated by the distribution heterogeneity, we propose *MixEmo*, a cross-subject emotion recognition framework tailored for mixed-distribution settings. *MixEmo* effectively enhances and models heterogeneous subsets while avoiding reliance on modality-specific backbone designs, making it readily adaptable to diverse emotional modalities.

- **Superior performance.** We conduct extensive experiments on five emotion datasets spanning three modalities–EEG signals (3D tensors), physiological indicators (high-dimensional vectors), and societal behaviors (multivariate time series). The results demonstrate the superiority of *MixEmo*. Moreover, we evaluate the contribution of each component and analyze the framework's sensitivity to key hyperparameters.

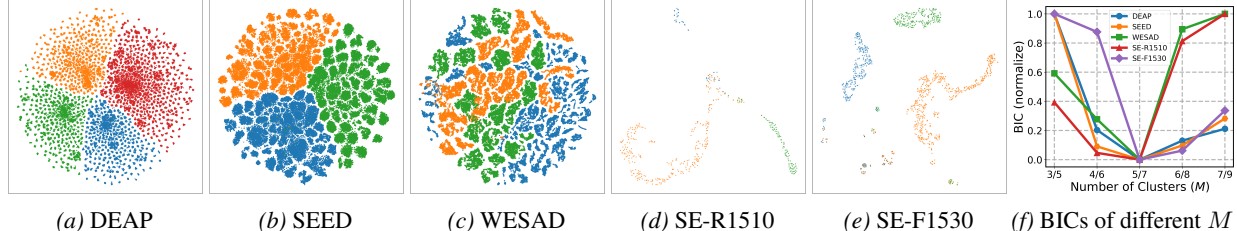

| *(a)* DEAP | *(b)* SEED | *(c)* WESAD | *(d)* SE-R1510 | *(e)* SE-F1530 | *(f)* BICs of different $M$ |

*Figure 2.* Visualization of representation distributions on five emotion recognition datasets, where different colors denote different emotion categories. As can be observed, the representations form multiple distinct clusters in the embedding space, indicating that a single dataset typically contains multiple heterogeneous distributions. Furthermore, when determining the optimal number of clusters, we employ the Bayesian Information Criterion (BIC) as a rigorous quantitative metric, where a lower BIC value indicates better clustering performance.

## 2. Preliminary

This section introduces the definition, motivation, and theoretical basis of the proposed framework.

### 2.1. Problem Statement

Given an emotion dataset $\mathcal{D} = \{(x_i, y_i)\}_{i=1}^N$, where $x$ denotes a sample–such as EEG signals, physiological signals, or other emotion modalities–and $y$ denotes its label, we simulate the cross-subject setting by partitioning the dataset into a training set $\mathcal{D}_{train}$ and a test set $\mathcal{D}_{test}$ based on subject identities. Specifically, we enforce $\mathcal{D}_{train} \bigcap \mathcal{D}_{test} = \emptyset$, meaning that the subjects in the training and test sets are completely non-overlapping. The model is trained on $\mathcal{D}_{train}$ and evaluated on $\mathcal{D}_{test}$, enabling an assessment of its cross-subject emotion recognition performance.

### 2.2. Motivation

To investigate the distributional characteristics of samples in the embedding space, we conducted experiments on five emotion datasets spanning three different modalities. Specifically, for each dataset, we construct a backbone with an encoder–classifier architecture. The backbone is first trained to achieve strong performance. Then, we extract sample representations using the encoder and visualize them using t-SNE. The experimental results are shown in Figure 2. The distributional heterogeneity of the datasets makes it difficult for unified modeling approaches to effectively capture multiple distributional subsets simultaneously, which may in turn limit the model's overall generalization performance.

### 2.3. Theoretical Analysis

The designs of both the UDG and HDCL are grounded in the following propositions, whose complete proofs provided in Appendix A.

**Proposition 2.1.** *Given two datasets $\mathcal{D}_1 \sim \mathcal{N}(\mu_1, \sigma_1^2)$ and $\mathcal{D}_2 \sim \mathcal{N}(\mu_2, \sigma_2^2)$, where $\mathcal{N}$ denotes the underlying distribution of the dataset, $\mu$ and $\sigma^2$ represent the mean and variance, respectively, we assume that $\mathcal{D}_1$ and $\mathcal{D}_2$ are*

*independent and satisfy $\mu_1 \neq \mu_2$ and $\sigma_1^2 \neq \sigma_2^2$. Suppose we randomly sample $m$ instances from $\mathcal{D}_1$ and $n$ instances from $\mathcal{D}_2$ to construct a new dataset $\mathcal{D}_3$:*

$$\mathcal{D}_3 = \left\{ \binom{\mathcal{D}_1}{m} \bigcup \binom{\mathcal{D}_2}{n} \right\} \sim \mathcal{N}(\mu_3, \sigma_3^2) \qquad (1)$$

*It follows that $\mu_3 \notin \{\mu_1, \mu_2\}$ and $\sigma_3^2 \notin \{\sigma_1^2, \sigma_2^2\}$, indicating that the synthesized dataset is distributionally heterogeneous with respect to the existing datasets.*

Proposition 2.1 demonstrates that it is possible to construct unseen distributions through random combinations of samples drawn from known distributions. These synthesized distributions are guaranteed to differ from the originals, thus increasing distributional diversity.

**Proposition 2.2.** *Given $M$ heterogeneous subsets $\{\mathcal{D}_i\}_{i=1}^M$, each subset $\mathcal{D}_i$ is randomly divided into a support set $\mathcal{D}_i^s$ and a query set $\mathcal{D}_i^q$. Let $\phi_g(\cdot; \theta_g)$ denote a globally shared prediction model, and let $\phi_i(\cdot; \theta_i)$ denote the proxy model associated with the $i$-th subset. Here, $\theta_g$ and $\theta_i$ represent the model parameters, and all models share the same network architecture. For each $\phi_i$, before every training iteration, its parameters are initialized with $\theta_g$. $\phi_i$ is then updated using gradient descent on $\mathcal{D}_i^s$, after which the query loss $\mathcal{L}_i^q$ is computed on $\mathcal{D}_i^q$. The parameters $\theta_g$ are subsequently updated using the aggregated query losses across all subsets:*

$$\theta_g = \theta_g - \alpha \frac{1}{M} \sum_{i=1}^M \nabla_{\theta_g} \mathcal{L}_i^q \qquad (2)$$

*where $\alpha$ is the learning rate and $\nabla_{\theta_g} \mathcal{L}_i^q$ denotes the gradient of the query loss with respect to $\theta_g$. Under this update rule, $\phi_g$ minimizes the expected query loss across all heterogeneous subsets.*

Proposition 2.2 indicates that the proposed HDCL enables learning a model that is jointly adapted to multiple heterogeneous subsets, thereby improving overall performance under mixed-distribution scenarios.

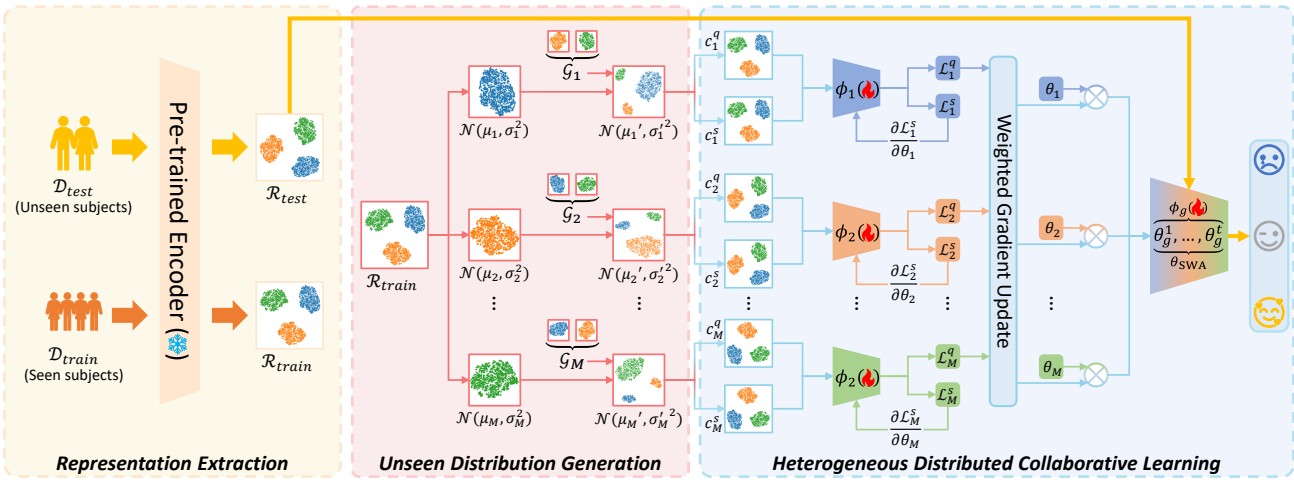

*Figure 3.* Overall of *MixEmo*. We first extract sample representations using a pretrained encoder. Next, we partition the sample representations into several single-distribution subsets, which serve as distribution prototypes. To enhance distributional diversity, we iteratively select one prototype $\mathcal{N}(\mu_i, \sigma_i^2)$ as the main subset and combine it with randomly sampled representations from other subsets $\mathcal{G}_i$, thereby constructing unseen-distribution subsets $(\mathcal{N}_i \cup \mathcal{G}_i)$. Finally, each subset $c_i$ is randomly divided into a support set $c_i^s$ and a query set $c_i^q$, which are used for training and evaluating the corresponding proxy model $\phi_i$, respectively. The query loss $\mathcal{L}_i^q$ is used as the weighting signal to optimize the globally shared prediction model $\phi_g$.

## 3. Methodology

*MixEmo* mainly consists of the UDG and the HDCL. The UDG generates unseen distributions to enhance distributional diversity, while the HDCL is designed to improve the model's learning capability and generalization performance on mixed-distribution datasets. The overall architecture of *MixEmo* is illustrated in Figure 3.

### 3.1. Representation Extraction

The backbone adopts an encoder–classifier architecture. To achieve the goal of weakly modality-dependent, we adopt widely used neural network architectures as encoders (as shown in Table 4) and employ a Multi-Layer Perceptron (MLP) as the classifier. The backbone is trained on $\mathcal{D}_{train}$, where its parameters are optimized by minimizing the cross-entropy loss between the predicted and ground-truth labels, yielding a well-performing encoder. Subsequently, We freeze the encoder parameters and transform $\mathcal{D}_{train}$ into the corresponding representation set $\mathcal{R}_{train}$ for subsequent training, while simultaneously converting $\mathcal{D}_{test}$ into $\mathcal{R}_{test}$ for later testing.

### 3.2. Unseen Distribution Generation

To enhance the distributional diversity of the data, we develop the UDG based on Proposition 2.1. As illustrated in Figure 2, a mixture-distribution dataset typically consists of several subsets that follow distinct underlying distributions. To separate these heterogeneous components from the dataset, we employ a Gaussian Mixture Model (GMM) to partition the representation set $\mathcal{R}_{train}$ into $M$ subsets,

each approximating a single distribution:

$$\mathcal{C} = \{c_i | c_i \sim \mathcal{N}(\mu_i, \sigma_i^2)\}_{i=1}^M \tag{3}$$

For any two subsets $c_i$ and $c_j$, we have:

$$\mu_i \neq \mu_j, \qquad \sigma_i^2 \neq \sigma_j^2 \tag{4}$$

Therefore, each subset can be regarded as an independent distribution prototype. We then iteratively treat each distribution prototype as the main subset, from which we randomly sample $m$ representations. For each of the remaining subsets, we randomly sample $n$ representations and combine them with the representations from the main subset to construct a new representation set:

$$c_i' = \binom{c_i}{m} \cup \left\{ \binom{c_j}{n} \right\}_{j=1, j \neq i}^M, \qquad \text{s.t.} \frac{m}{n} \geq M - 1 \tag{5}$$

This constraint requires the sampling ratio $(m/n)$ to be no less than $M - 1$, ensuring that samples from the main subset dominate each synthesized subset. As a result, each synthesized subset is globally biased toward the main subset distribution, thereby guaranteeing the diversity of the synthesized distributions. Extending Proposition 2.1 to the multi-subset setting yields:

$$\mu_i' \notin \{\mu_i\}_{i=1}^M, \qquad \sigma_i'^2 \notin \{\sigma_i^2\}_{i=1}^M \tag{6}$$

Therefore, the UDG can effectively generate representation sets that follow unseen distributions, thereby improving the overall distributional diversity of the data.

**Optimal M Search.** To determine a reasonable range for the number of clusters $M$, we adopt the BIC (Schwarz, 1978) as

the metric for evaluating clustering quality. We first define a search interval $[s, e]$ and a step size $\tau$. Leveraging the empirical observation that BIC typically exhibits a concave trend with respect to the number of clusters, we evaluate the BIC values at increments of $\tau$ within this interval and select the cluster number corresponding to the minimum BIC as the current optimal value $M^*$. The search interval is then updated to $[(s + M^*)/2, (e + M^*)/2]$, and the step size is reduced to $\tau/2$. This procedure is repeated until $\tau = 1$. The final $M^*$ is taken as the optimal number of clusters. The algorithmic details are provided in Appendix B.1.

### 3.3. Heterogeneous Distributed Collaborative Learning

To enable collaborative learning across multiple heterogeneous subsets, we design the HDCL based on Proposition 2.2. To facilitate parameter sharing and joint optimization among subsets, we introduce a globally shared prediction model $\phi_g(\cdot; \theta_g)$ and construct a proxy model $\phi_i(\cdot; \theta_i)$ for each subset $c_i$, where $\phi_i$ shares the same architecture as $\phi_g$, and all models are initially parameterized by $\theta_g$.

The training of the HDCL consists of two alternating stages. For each subset $c_i$, we randomly divide it into a support set $c_i^s$ and a query set $c_i^q$, which are used to updating the parameters of $\phi_i$ and $\phi_g$, respectively. During the proxy training stage, we sequentially train each $\phi_i$ using its corresponding $c_i^s$. The parameters are optimized via the cross-entropy loss between the predicted outputs and the ground-truth:

$$\mathcal{L}_i^s = -\frac{1}{|c_i^s|} \sum_{(r_j, y_j) \in c_i^s} y_j \log\left(\phi_i(r_j; \theta_j)\right) \quad (7)$$

After proxy training, the HDCL aggregates the parameters of all proxy models. the performance of a proxy model on its support set is not considered fully reliable; instead, its contribution should be measured based on its inference performance on the corresponding query set. Specifically, for each $\phi_i$, we compute the corresponding query loss $\mathcal{L}_i^q$ on $c_i^q$. Subsequently, $\phi_g$ is then optimized to minimize the expected query loss across all heterogeneous subsets:

$$\mathcal{L}_{HDCL} = \min_{\theta_g} \frac{1}{M} \sum_{i=1}^{M} \mathcal{L}_i^q \quad (8)$$

According to Proposition 2.2, the parameter update of $\phi_g$ can be expressed as:

$$\theta_g = \theta_g - \alpha \frac{1}{M} \sum_{i=1}^{M} \nabla_{\theta_g} \mathcal{L}_i^q \quad (9)$$

To accelerate the convergence of the HDCL, we propose a Weighted Gradient Update (WGU) to replace simple averaging. Specifically, we quantify the importance of each subset to $\phi_g$ based on the magnitude of its query loss, and replace

the uniform weight $1/M$ with a subset-specific importance coefficient $\omega_i$, which is computed as:

$$\omega_i = \frac{\exp\left(\mathcal{L}_i^q\right)}{\sum_{j=1}^{M} \exp\left(\mathcal{L}_j^q\right)} \quad (10)$$

Accordingly, the parameter update of $\phi_g$ becomes:

$$\theta_g = \theta_g - \alpha \sum_{i=1}^{M} \omega_i \nabla_{\theta_g} \mathcal{L}_i^q \quad (11)$$

Due to the substantial distributional discrepancies among different subsets, even when the training loss continues to decrease, the generalization performance of $\phi_g$ on the test set cannot be reliably guaranteed. In such cases, the model is prone to converging to certain sharp local optima, which leads to unstable performance. To mitigate this issue, we introduce Stochastic Weight Averaging (SWA) to guide the model toward a flatter optimum, denoted as $\theta_{\text{SWA}}$. The update of $\theta_{\text{SWA}}$ aggregates the parameters $\theta_g$ obtained after each round of alternating training. Using $t$ to represent the current number of iterations, its update formula is:

$$\theta_{\text{SWA}}^{t+1} = \frac{\theta_{\text{SWA}}^t + \theta_g^t}{t + 1} \quad (12)$$

Through multiple rounds of alternating training combined with the SWA, $\phi_g(\cdot; \theta_{\text{SWA}})$ is able to effectively integrate knowledge from heterogeneous subsets and approximate their joint optimal solution. As a result, the proposed approach achieves more stable and superior generalization performance under mixed-distribution scenarios. A complexity analysis of the HDCL is provided in Appendix B.2. At this point, one round of alternating training in HDCL is complete. Before proceeding to the next round, we set $\{\theta_i\}_{i=1}^{M} = \theta_g$, thereby establishing parameter coupling between $\phi_g$ and the subset-specific models $\{\phi_i\}_{i=1}^{M}$.

**Convergence.** Since we set $\{\theta_i\}_{i=1}^{M} = \theta_g$ at the beginning of each training iteration, $\mathcal{L}_{HDCL}$ can be regarded as a composite function of the "proxy model update mapping" and the "query loss".

**Theorem 3.1.** *Assume that the support-set loss admits smooth descent, the stochastic gradients are bounded with controlled variance, and $\alpha$ is smaller than a certain constant. Then, there exists a constant $C > 0$ such that*

$$\frac{1}{T} \sum_{t=0}^{T-1} \mathbb{E}\left[\|\nabla_\theta \mathcal{L}_{HDCL}(\theta_t)\|^2\right] \leq \frac{C}{T} \quad (13)$$

where $T$ denotes the number of training iterations. The detailed proof of this theorem is provided in Appendix A.3. This result indicates that, under the above standard optimization assumptions, the composite objective function associated with the HDCL preserves controlled smoothness, thereby guaranteeing favorable convergence properties for the HDCL optimization process.

## 3.4. Comparison and Discussion

**Comparison.** Most existing cross-subject emotion recognition methods are primarily designed for EEG signals, making them difficult to transfer to other emotional modalities. In contrast, *MixEmo* adopts a general neural network as the encoder, endowing it with stronger modality extensibility. Among existing approaches, Mixup (Zhang et al., 2017) constructs unseen data by linearly combining samples at the instance level. However, for emotion data with strong structural constraints, this strategy may easily introduce semantic conflicts. By contrast, the UDG constructs unseen distributions at the distribution level, thereby effectively avoiding conflicts. Model-Agnostic Meta-Learning (MAML) (Finn et al., 2017) represent a typical paradigm for collaborative learning, but their performance often degrades in scenarios with pronounced heterogeneity across data distributions. In comparison, the HDCL integrates WGU with SWA to guide the model toward stable optimization across multiple heterogeneous subsets, thereby promoting convergence while enhancing the model's generalization capability.

**Limitation.** One of the core ideas of *MixEmo* is to enhance distributional diversity to fill sparse regions in the embedding space. However, when representations exhibit highly compact intra-class structures and well-separated inter-class boundaries (as illustrated in Figure 2a, DEAP dataset), the generated distributions may only partially cover these sparse regions. Nevertheless, given *MixEmo*'s substantial advantages in weak modality dependency and heterogeneous distribution modeling, we believe that this direction represents a promising avenue for future research in the field.

# 4. Experiment

This section introduces the datasets, baselines, evaluation metrics, and experimental configurations employed to assess the performance of *MixEmo*. The implementation details are provided in Appendix C. Code is publicly available [1].

## 4.1. Experimental Setup

### 4.1.1. DATASET

We conduct experiments on five datasets, which are summarized in Table 1. These datasets cover three distinct emotion modalities. Specifically, DEAP (Koelstra et al., 2012) and SEED (Zheng & Lu, 2015) are EEG signal datasets represented as 3D tensors; WESAD (Schmidt et al., 2018) is a physiological indicator dataset represented as high-dimensional vectors; and SE-R1510 (Chen, 2023) and SE-F1530 (Chen, 2023) are societal behavior datasets represented as multivariate time series. Details of these datasets are provided in Appendix D.

---

[1] https://github.com/WendingXiong/MixEmo

*Table 1.* Detailed information of the datasets.

| Dataset | #Item | #Subject | #Label | Modality |
|---------|-------|----------|--------|----------|
| DEAP | 76,800 | 32 | 4 | EEG signal |
| SEED | 152,730 | 15 | 3 | EEG signal |
| WESAD | 4,641,279 | 15 | 3 | Physiological signal |
| SE-R1510 | 714,587 | 15 | 3 | Societal behavior |
| SE-F1530 | 1,815,531 | 15 | 3 | Societal behavior |

### 4.1.2. BASELINE

To comprehensively evaluate the performance of *MixEmo*, we compare it against a variety of state-of-the-art data augmentation methods, including Mixup (Zhang et al., 2017), MAML (Finn et al., 2017), CTGAN (Xu et al., 2019), GT-GAN (Jeon et al., 2022), PCF-GAN (Lou et al., 2023), InfoTS (Luo et al., 2023), and TimeDP (Huang et al., 2025). Among them, CTGAN, GT-GAN, and PCF-GAN are GAN-based approaches, InfoTS represents an information-aware method, and TimeDP is built upon the diffusion-model framework. In addition, we include several cross-subject emotion recognition methods designed specifically for EEG signals as baselines, including EEGMatch (Zhou et al., 2025), DMMR (Wang et al., 2024), MoGE (Liu et al., 2024c), and EmT(Ding et al., 2025). Details of these baselines are provided in Appendix E.

### 4.1.3. EVALUATION METRICS

We adopt four metrics to evaluate all methods, namely Accuracy, Precision, Recall, and F1-score (macro).

### 4.1.4. RESEARCH QUESTION

This subsection presents the experiments conducted to demonstrate the effectiveness of *MixEmo*. In the reported results, **bold** denotes the best performance and underline indicates the second-best. For each dataset, cross-validation is employed for evaluation. To comprehensively evaluate *MixEmo*, we organize our analysis around the following research questions:

**Superiority**–Does *MixEmo* outperform state-of-the-art data augmentation methods and EEG-specific cross-subject emotion recognition methods? (Section 4.2.1 and Section 4.2.2)

**Ablation Study**–How does each component contribute to the overall performance of *MixEmo*? (Section 4.3)

**Sensitivity Analysis**–How sensitive is *MixEmo* to different hyperparameter settings? (Section 4.4)

**Convergence Analysis**–How different learning rates and weight update strategies affect the convergence behavior of *MixEmo*? (Section 4.5)

**Case Study**–Why does *MixEmo* surpass existing data augmentation methods in performance? (Section 4.6)

*Table 2.* Performance comparison between *MixEmo* and state-of-the-art data augmentation methods. For intuitive comparison, the table also reports the backbone performance under non-cross-subject settings (😄)–where the training and test sets include the same subjects–and cross-subject settings (🥴), with all values presented in percentage (%).

| Dataset | DEAP | | SEED | | WESAD | | SE-R1510 | | SE-F1530 | |
|---|---|---|---|---|---|---|---|---|---|---|
| Method | Acc | F1 | Acc | F1 | Acc | F1 | Acc | F1 | Acc | F1 |
| Backbone (😄) | 88.25 | 88.27 | 68.49 | 68.49 | 97.01 | 97.02 | 69.70 | 63.05 | 60.67 | 55.32 |
| Backbone (🥴) | 25.15 | 25.03 | 44.26 | 44.54 | 52.34 | 50.31 | 48.91 | 27.34 | 30.23 | 27.87 |
| Mixup | 27.32 | 23.44 | 51.20 | 50.82 | 43.17 | 41.80 | 57.55 | 44.13 | 46.63 | 39.06 |
| MAML | 25.04 | 24.73 | 47.22 | 46.63 | 48.41 | 40.56 | 44.57 | 25.56 | 48.09 | 42.42 |
| CTGAN | 26.08 | 25.25 | 58.52 | 57.40 | 66.59 | 64.52 | 58.95 | 47.95 | 49.62 | 44.70 |
| GT-GAN | 25.25 | 24.44 | 57.78 | 56.07 | 67.59 | 65.66 | 55.88 | 44.54 | 47.88 | 41.14 |
| PCF-GAN | 26.29 | 25.55 | 55.19 | 53.76 | 61.56 | 61.47 | 57.30 | 38.52 | 44.44 | 39.60 |
| InfoTS | 25.58 | 25.32 | 47.04 | 47.08 | 47.26 | 47.54 | 53.68 | 32.41 | 43.37 | 39.69 |
| TimeDP | 25.20 | 23.99 | 48.33 | 47.71 | 55.89 | 54.34 | 54.21 | 28.21 | 39.31 | 33.47 |
| *MixEmo* | **29.52** | **30.34** | **62.04** | **60.32** | **70.37** | **67.58** | **62.50** | **48.68** | **52.67** | **45.80** |

## 4.2. Superiority

### 4.2.1. MIXEMO VS. GENERAL METHODS

This subsection compares the performance of *MixEmo* with state-of-the-art data augmentation methods, and the results are summarized in Table 2. The findings indicate that: 1) *MixEmo* significantly enhances model generalization. On the SE-F1530 and WESAD datasets, the maximum improvements in accuracy and F1-score reach 22.44% and 17.27%, respectively, demonstrating that the *MixEmo* effectively strengthens model robustness on unseen subjects. 2) *MixEmo* performs consistently across different types of datasets. Even on the relatively low-heterogeneity SEED dataset, *MixEmo* achieves the best performance, highlighting its broad applicability and robustness across datasets with varying distribution structures.

### 4.2.2. MIXEMO VS. EEG-SPECIFIC METHODS

This subsection compares *MixEmo* with state-of-the-art cross-subject emotion recognition methods specifically designed for EEG signals. Since the architectures and feature extraction techniques of these baselines are tailored to the EEG modality, this comparison is conducted only on EEG datasets, and the results are presented in Table 3. Although *MixEmo* does not consistently outperform these EEG-specific cross-subject methods, the limitations of the baselines are evident–they are difficult to generalize to other emotion modalities. Given the rapid development of multimodal emotion recognition research, we argue that *MixEmo* remains a highly competitive approach.

## 4.3. Ablation Study

This subsection analyzes the contributions of different components to *MixEmo*'s performance. In the ablation of HDCL, the multiple heterogeneous subsets input to HDCL are merged into a single set, thereby removing the model's

*Table 3.* Performance comparison between *MixEmo* and state-of-the-art EEG-specific cross-subject emotion recognition methods on the DEAP and SEED datasets. All methods take raw EEG signals as input (%).

| Dataset | DEAP | | SEED | |
|---|---|---|---|---|
| Method | Acc | F1 | Acc | F1 |
| EEGMatch | 27.63 | 26.47 | 60.56 | 58.98 |
| DMMR | 27.42 | 27.13 | **62.41** | **61.35** |
| MoGE | 27.28 | 26.23 | 61.30 | 60.22 |
| EmT | 26.83 | 26.63 | 59.26 | 57.87 |
| *MixEmo* | **29.52** | **30.34** | 62.04 | 60.32 |

ability to perform collaborative optimization across subsets. The experimental results are shown in Figure 4. The findings indicate that: 1) Removing either UDG or HDCL leads to a significant drop in overall performance. Specifically, UDG effectively enhances the distributional diversity of the training data by generating unseen distributions, while HDCL substantially improves the model's generalization ability under heterogeneous distributions. 2) WGU primarily facilitates stable convergence during training, exerting a relatively limited impact on final performance. In contrast, SWA guides the model toward flatter optima, effectively mitigating overfitting and contributing more significantly to the overall performance improvement.

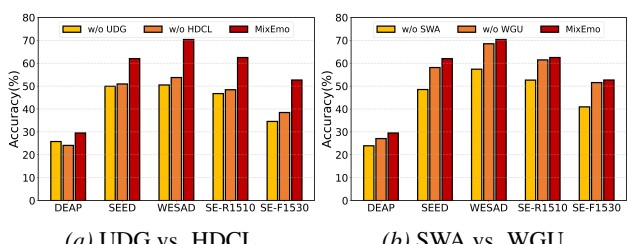

*(a)* UDG vs. HDCL      *(b)* SWA vs. WGU

*Figure 4.* Ablation study results on five datasets. "w/o" indicates the removal of the corresponding component.

### 4.4. Sensitivity Analysis

This subsection analyzes the sensitivity of *MixEmo* to the representation dimension $d$, the number of subsets $M$, and the sampling ratio $m/n$. Regarding $M$, although the theoretically optimal subset number $M^*$ can be obtained via the BIC criterion, we further investigate its effect on *MixEmo*'s performance. To this end, we evaluate $M^*$ and several neighboring values, with the results shown in Figure 5. The findings indicate that: 1) *MixEmo* is relatively sensitive to changes in $M$, so it is advisable to search around the BIC-provided reference value. Notably, choosing a slightly larger $M$ does not significantly degrade performance, though it incurs additional computational cost; 2) Optimal performance is achieved at $d = 512$ and $d = 1024$; a too-small dimension may limit the model's representational capacity, while a too-large dimension may lead to overfitting; 3) The ratio $m/n$ determines the number of samples in the synthesized subsets. When a distribution prototype has limited samples, the corresponding synthesized subset will also be small, potentially affecting HDCL's learning effectiveness. Therefore, it is recommended to set $m/n$ to its maximum value while satisfying the imposed constraints.

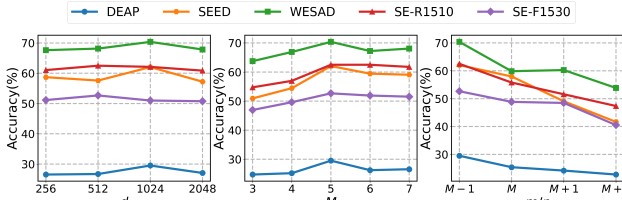

*Figure 5.* Sensitivity analysis with respect to the representation dimension $d$, the number of subsets $M$, and the sampling ratio $m/n$.

### 4.5. Convergence Analysis

This subsection analyzes the effects of the learning rate $\alpha$ and the weight update strategies on the convergence of *MixEmo*. Experiments were conducted on the SE-F1530 and SE-R1510 datasets. For ease of comparison and visualization, all results were normalized, as shown in Figures 6 and 7. The findings indicate that: 1) an excessively large learning rate may hinder stable convergence, whereas a very small learning rate ensures smooth training loss descent but generally requires more iterations to reach optimal performance; 2) compared with the simple average weight update strategy, WGU facilitates faster and more stable convergence, while SWA primarily enhances the model's generalization ability, contributing little to convergence speed.

### 4.6. Case Study

The core idea of generative data augmentation is to enhance model robustness by enriching the diversity of training samples and filling the gaps in the embedding space. To analyze why *MixEmo* achieves superior performance, we compare its augmentation effects with two representative approaches—CTGAN and TimeDP—on the WESAD dataset. The results are shown in Figure 8.

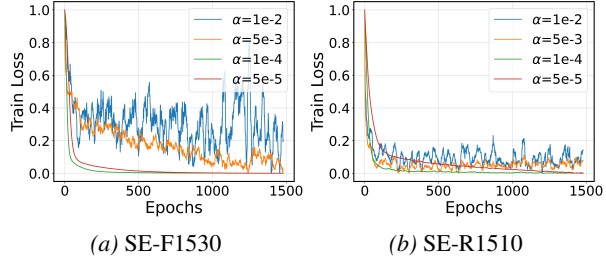

|  |  |
|---|---|
| *(a)* SE-F1530 | *(b)* SE-R1510 |

*Figure 6.* Convergence behavior under different values of $\alpha$.

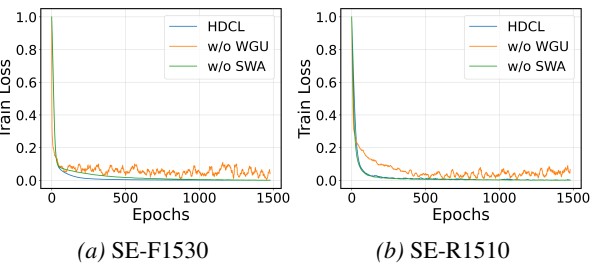

|  |  |
|---|---|
| *(a)* SE-F1530 | *(b)* SE-R1510 |

*Figure 7.* Convergence behavior under different weight update strategies.

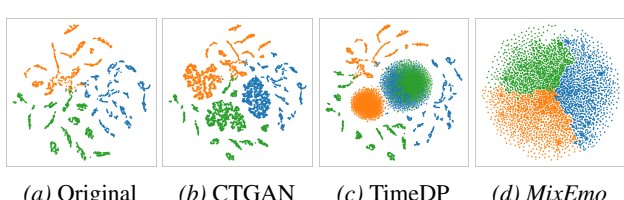

| *(a)* Original | *(b)* CTGAN | *(c)* TimeDP | *(d) MixEmo* |
|---|---|---|---|

*Figure 8.* Comparison of augmentation effectiveness. For illustrative purposes, we randomly selected 6,000 sample representations from the WESAD training set (2,000 per class) to construct the base training set. Each augmentation method generated synthetic representations based on these samples, and we visualized the distributions of both real and generated representations via t-SNE.

The experimental results show that TimeDP not only exhibits limited diversity when generating representations for different labels, but also fails to maintain clear class separability. CTGAN achieves slightly better separation, yet the diversity of its generated samples remains limited, revealing its shortcomings in mixed-distribution scenarios. In contrast, *MixEmo* excels in both diversity and discriminability. Its generated representations effectively fill the distributional gaps of the original samples in the embedding space, thereby substantially reducing the risk of encountering unseen samples during inference.

## 5. Conclusion

This study focuses on the problem of cross-subject emotion recognition. Given that existing approaches are often tightly coupled with specific emotion modalities and seldom account for mixed-distribution scenarios, we first uncover the presence of distributional heterogeneity within emotion datasets. This observation motivates us to enhance distributional diversity and explicitly model the heterogeneous subsets within the data. Building on these insights, we propose *MixEmo*, a weakly modality-dependent cross-subject emotion recognition framework designed to improve model generalization under mixed-distribution conditions. *MixEmo* employs the UDG and HDCL to enhance distributional diversity and enable collaborative learning across heterogeneous subsets, respectively. Extensive experimental results demonstrate that *MixEmo* significantly outperforms state-of-the-art data augmentation approaches as well as EEG-specific cross-subject emotion recognition methods.

## Acknowledgment

This work is supported by the National Natural Science Foundation of China (Grant Nos. U22A2035, U25A7007, 62572359, 62506308), the Chengdu Science and Technology Program (Grant No. 2025-YF12-00030-RC), the Natural Science Foundation of Xinjiang Uygur Autonomous Region (Grant No. 2025D01C293), the Open Project of Hubei Provincial Key Laboratory of Multimedia Network Communication Engineering (Grant No. 2025KFKT14), the Hubei Provincial Teaching Reform Research Project (Grant No. 2025265), and the Hubei Higher Education Society Education Research Project (Grant No. 2024XD198).

## Impact Statement

(1) **Data bias and generalization risks**. Emotion datasets may contain biases related to demographics, cultural backgrounds, or specific populations, which may limit generalization and lead to degraded performance or systematic bias for underrepresented groups. (2) **Potential misuse in sensitive scenarios**. Emotion recognition technologies may be applied in high-risk contexts such as surveillance, profiling, or automated decision-making. Without proper ethical oversight, transparency, and informed consent, such applications may raise privacy concerns and fairness issues. We will explicitly state that this work should not be deployed in high-risk or sensitive scenarios without strict ethical and regulatory safeguards.

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

# A. Proof Details

## A.1. Proof of Proposition 2.1

*Proof.* If $\mathcal{D}_1 \sim \mathcal{N}(\mu_1, \sigma_1{}^2)$ and $\mathcal{D}_2 \sim \mathcal{N}(\mu_2, \sigma_2{}^2)$. Let $m$ and $n$ samples be randomly drawn from $\mathcal{D}_1$ and $\mathcal{D}_2$, respectively, to form a new dataset $\mathcal{D}_3$. Based on the definition of the sample mean, the expression for the sample mean of $\mathcal{D}_3$ can be derived as follows:

$$\mu_3 = \frac{\sum\limits_{i=1}^{m} x_{1,i} + \sum\limits_{i=1}^{n} x_{2,i}}{m+n} = \frac{m\mu_1 + n\mu_2}{m+n} \tag{14}$$

Based on the definition of variance, the expression for the sample variance of $\mathcal{D}_3$ can be formulated as follows:

$$\sigma_3{}^2 = \frac{1}{(m+n)} \Big( \underbrace{\sum_{i=1}^{m} (x_{1,i} - \mu_3)^2}_{P_1} + \underbrace{\sum_{i=1}^{n} (x_{2,i} - \mu_3)^2}_{P_2} \Big) \tag{15}$$

By expanding $P_1$, we obtain:

$$\begin{aligned} P_1 &= \sum_{i=1}^{m} \Big[ \big( (x_{1,i} - \underline{\mu_1}) + (\underline{\mu_1} - \mu_3) \big)^2 \Big] \\ &= \sum_{i=1}^{m} (x_{1,i} - \mu_1)^2 + 2(\mu_1 - \mu_3) \sum_{i=1}^{m} (x_{1,i} - \mu_1) + \sum_{i=1}^{m} (\mu_1 - \mu_3)^2 \\ &= m\sigma_1^2 + 2(\mu_1 - \mu_3) \sum_{i=1}^{m} (x_{1,i} - \mu_1) + m(\mu_1 - \mu_3)^2 \end{aligned} \tag{16}$$

Since $\sum_{i=1}^{m} (x_{1,i} - \mu_1) = 0$, it follows that:

$$\begin{aligned} P_1 &= m\sigma_1^2 + m(\mu_1 - \mu_3)^2 \\ &= m\sigma_1^2 + m\left( \mu_1 - \frac{m\mu_1 + n\mu_2}{m+n} \right)^2 \\ &= m\sigma_1^2 + m\left( \frac{n}{m+n} \right)^2 (\mu_1 - \mu_2)^2 \end{aligned} \tag{17}$$

Similarly, by expanding $P_2$, we obtain:

$$P_2 = n\sigma_2^2 + n\left( \frac{m}{m+n} \right)^2 (\mu_1 - \mu_2)^2 \tag{18}$$

By substituting $P_1$ and $P_2$ back into the Equation 15, we obtain:

$$\begin{aligned} \sigma_3{}^2 &= \frac{m\sigma_1^2 + m\left(\frac{n}{m+n}\right)^2 (\mu_1 - \mu_2)^2 + n\sigma_2^2 + n\left(\frac{m}{m+n}\right)^2 (\mu_1 - \mu_2)^2}{m+n} \\ &= \frac{m\sigma_1{}^2 + n\sigma_2{}^2}{m+n} + \frac{mn(\mu_1 - \mu_2)^2}{(m+n)^2} \end{aligned} \tag{19}$$

Evidently, $\mu_3 = \mu_1$ and $\sigma_3{}^2 = \sigma_1{}^2$ and if and only if $n = 0$; similarly, $\mu_3 = \mu_2$ and $\sigma_3{}^2 = \sigma_2{}^2$ and if $m = 0$. Based on this analysis, Proposition 2.1 holds. $\qquad\square$

## A.2. Proof of Proposition 2.2

*Proof.* Given $M$ heterogeneous subsets $\{\mathcal{D}_i\}_{i=1}^{M}$, we begin by randomly dividing each subset into a support set $\mathcal{D}_i^s$ and a query set $\mathcal{D}_i^q$, and collaboratively learn across all subsets using $M$ proxy models $\{\phi_i(\cdot; \theta_i)\}_{i=1}^{M}$ and a globally shared prediction model $\phi_g(\cdot; \theta_g)$. At the beginning of each training round, we initialize all proxy models with the prediction

model's parameters, i.e., $\{\theta_i\}_{i=1}^{M} = \theta_g$. If we omit the importance weights $\omega$, the update rule for the parameters $\theta_i$ of the proxy model $\phi_i$, trained on its corresponding support set, is given by:

$$\theta_i = \theta_i - \alpha \nabla_{\theta_i} \mathcal{L}_i^s(\theta_i) = \theta_g - \alpha \nabla_{\theta_g} \mathcal{L}_i^s(\theta_g) \tag{20}$$

where $\alpha$ denotes the learning rate and $\mathcal{L}_i^s(\theta_i)$ is the loss of $\phi_i$ evaluated on $\mathcal{D}_i^s$. Our objective is to optimize $\phi_g$ such that the expected loss on each query set is minimized. Based on the Equation 20, the objective function of HDCL can be formulated as follows:

$$\mathcal{L}_{HDCL} = \min_{\theta_g} \frac{1}{M} \sum_{i=1}^{M} \mathcal{L}_i^q(\theta_i) = \min_{\theta_g} \frac{1}{M} \sum_{i=1}^{M} \mathcal{L}_i^q \big( \theta_g - \alpha \nabla_{\theta_g} \mathcal{L}_i^s(\theta_g) \big) \tag{21}$$

where $\mathcal{L}_i^q(\theta_i)$ denotes the loss of $\phi_i$ on $\mathcal{D}_i^q$. Therefore, $\mathcal{L}_{HDCL}$ is a composite function with respect to $\theta_g$. Taking the gradient of $\mathcal{L}_{HDCL}$ with respect to $\theta_g$ and applying the chain rule, we obtain:

$$
\begin{aligned}
\nabla_{\theta_g} \mathcal{L}_{HDCL} &= \frac{1}{M} \sum_{i=1}^{M} \nabla_{\theta_g} \mathcal{L}_i^q(\theta_i) \\
&= \frac{1}{M} \sum_{i=1}^{M} \nabla_{\theta_i} \mathcal{L}_i^q(\theta_i) \cdot \frac{\partial \theta_i}{\partial \theta_g} \\
&= \frac{1}{M} \sum_{i=1}^{M} \nabla_{\theta_i} \mathcal{L}_i^q(\theta_i) \cdot \frac{\partial \big[ \theta_g - \alpha \nabla_{\theta_g} \mathcal{L}_i^s(\theta_g) \big]}{\partial \theta_g} \\
&= \frac{1}{M} \sum_{i=1}^{M} \nabla_{\theta_i} \mathcal{L}_i^q(\theta_i) \cdot \big( I - \alpha \nabla_{\theta_g}^2 \mathcal{L}_i^s(\theta_g) \big)
\end{aligned}
\tag{22}
$$

By neglecting the higher-order derivative terms $\nabla_{\theta_g}^2 \mathcal{L}_i^s(\theta_g)$, the gradient simplifies to:

$$\nabla_{\theta_g} \mathcal{L}_{HDCL} \approx \frac{1}{M} \sum_{i=1}^{M} \nabla_{\theta_i} \mathcal{L}_i^q(\theta_i) \tag{23}$$

Therefore, if the parameter update of $\phi_g$ is defined as:

$$\theta_g = \theta_g - \alpha \frac{1}{M} \sum_{i=1}^{M} \nabla_{\theta_g} \mathcal{L}_i^q \tag{24}$$

then the loss gradients on the query sets can be propagated to $\phi_g$. As the objective function is continuously optimized, $\phi_g$ converges toward minimizing the expected loss over all heterogeneous subsets. Consequently, Proposition 2.2 follows. $\square$

### A.3. Proof of Theorem 3.1

Our objective is to demonstrate that, under standard assumptions, the gradient norm of the prediction model $\phi_g$ converges when stochastic gradient descent is employed with an appropriately chosen learning rate $\alpha$.

### A.3.1. DEFINITION

**Definition A.1. (Loss Function):** Since both the support and query sets originate from the same homogeneous subset, we denote the corresponding support loss and query loss for subset $c_i$ uniformly as $\mathcal{L}_i$.

**Definition A.2. (Single-Step Gradient Descent):** Since all model parameters are synchronized to $\theta_g$ at the beginning of each training iteration, we uniformly use $\theta$ to denote the model parameters. Therefore, for any subset $c_i$, the single-step gradient descent update during the proxy training phase is:

$$\theta' = \theta - \alpha \nabla_\theta \mathcal{L}_i(\theta) \tag{25}$$

**Definition A.3. (Objective Function):** The HDCL aims to minimize the expected loss of the prediction model across all subsets $\mathcal{C}$ (where $|\mathcal{C}| = M$). Therefore, the objective function $F(\theta)$ is defined as:

$$F(\theta) = \mathbb{E}_{i \in [1, M]} \big[ \mathcal{L}_i(\theta') \big] = \mathbb{E}_{i \in [1, M]} \big[ \mathcal{L}_i \big( \theta - \alpha \nabla_\theta \mathcal{L}_i(\theta) \big) \big] \tag{26}$$

**Definition A.4. (Objective Gradient):** Differentiating $F(\theta)$ with respect to $\theta$, we denote the resulting gradient by $g_i(\theta)$. Applying the chain rule yields:

$$\nabla_\theta F(\theta) = \mathbb{E}_{i \in [1,M]}\big[\nabla_\theta \mathcal{L}_i(\theta')\big] = \mathbb{E}_{i \in [1,M]}\big[g_i(\theta)\big], \qquad g_i(\theta) = \nabla_\theta \mathcal{L}_i(\theta') = \nabla_{\theta'} \mathcal{L}_i(\theta') \cdot \big(I - \alpha \nabla_\theta^2 \mathcal{L}_i(\theta)\big) \tag{27}$$

**Definition A.5. (Aggregated Objective Gradient):** Let there be $M$ heterogeneous subsets. At iteration $t$, the aggregated objective gradient is:

$$\widehat{g}_t = \frac{1}{M} \sum_{i=1}^{M} g_i(\theta) \tag{28}$$

Therefore, the parameter update rule becomes:

$$\theta_{t+1} = \theta_t - \alpha \widehat{g}_t \tag{29}$$

### A.3.2. BASIC ASSUMPTIONS

**Assumption A.6. (Smoothness):** There exists a constant $L > 0$ such that, for any proxy model $\phi_i$, its loss function $\mathcal{L}_i$ is $L$-smooth during proxy training, i.e.,

$$\|\nabla_\theta \mathcal{L}_i(\theta) - \nabla_\theta \mathcal{L}_i(\theta')\| \leq L \|\theta - \theta'\|, \qquad \forall \theta \tag{30}$$

**Assumption A.7. ($\rho$-Lipschitz Hessian):** There exists a constant $\rho > 0$ such that the Hessian of $\mathcal{L}_i$ satisfies:

$$\left\|\nabla_\theta^2 \mathcal{L}_i(\theta) - \nabla_\theta^2 \mathcal{L}_i(\theta')\right\| \leq \rho \|\theta - \theta'\|, \qquad \forall \theta \tag{31}$$

**Assumption A.8. (Bounded Gradient Norm and Variance):** There exist constants $\mu > 0$ and $\sigma^2 > 0$ such that:

$$\mathbb{E}_{i \in [1,M]}\big[\|\nabla_\theta \mathcal{L}_i(\theta)\|^2\big] \leq \mu^2, \qquad \mathbb{E}_{i \in [1,M]}\big[\|g_i(\theta) - \nabla_\theta \mathcal{L}_i(\theta)\|^2\big] \leq \sigma^2, \qquad \forall \theta \tag{32}$$

**Assumption A.9. (Sufficiently Small Learning Rate):** The learning rate $\alpha$ is chosen small enough to satisfy the inequalities required in subsequent derivations. In general, it holds that:

$$\alpha \leq \frac{1}{2 L_F \sqrt{T}} \tag{33}$$

where $L_F$ denotes the smoothness constant of $F(\theta)$, and $T$ is the total number of training iterations.

### A.3.3. LEMMA PROOF

**Lemma A.10.** *(Lipschitz Continuity of the Objective Gradient): Based on Assumption A.6, the gradient of the objective function $\nabla_\theta F(\theta)$ is $L_F$-Lipschitz continuous, where $L_F = L(1 + \alpha L) + \alpha \rho \mu$.*

*Proof.* Fix subset index $i$. For any two parameters $\theta_1$ and $\theta_2$, their one-step gradient-descent updates are given by:

$$\theta_1' = \theta_1 - \alpha \nabla_{\theta_1} \mathcal{L}_i(\theta_1), \qquad \theta_2' = \theta_2 - \alpha \nabla_{\theta_2} \mathcal{L}_i(\theta_2) \tag{34}$$

The norm of the difference between their target gradients, by Definition A.4, is given by:

$$\|g_i(\theta_1) - g_i(\theta_2)\| = \left\|\nabla_{\theta_1'} \mathcal{L}_i(\theta_1') \cdot \big(I - \alpha \nabla_\theta^2 \mathcal{L}_i(\theta_1)\big) - \nabla_{\theta_2'} \mathcal{L}_i(\theta_2') \cdot \big(I - \alpha \nabla_\theta^2 \mathcal{L}_i(\theta_2)\big)\right\|$$
$$\leq \underbrace{\left\|\nabla_{\theta_1'} \mathcal{L}_i(\theta_1') - \nabla_{\theta_2'} \mathcal{L}_i(\theta_2')\right\|}_{P_1} + \alpha \underbrace{\left\|\nabla_{\theta_1'} \mathcal{L}_i(\theta_1') \cdot \alpha \nabla_\theta^2 \mathcal{L}_i(\theta_1) - \nabla_{\theta_2'} \mathcal{L}_i(\theta_2') \cdot \alpha \nabla_\theta^2 \mathcal{L}_i(\theta_2)\right\|}_{P_2} \tag{35}$$

By Assumption A.6, for $P_1$, we have:

$$
\begin{aligned}
P_1 &\leq L \|\theta_1' - \theta_2'\| \\
&\leq L \Bigg( \|\theta_1 - \theta 2\| + \alpha \underbrace{\|\nabla_{\theta_1} \mathcal{L}_i(\theta_1) - \nabla_{\theta_2} \mathcal{L}_i(\theta_2)\|}_{\leq L \|\theta_1 - \theta_2\|} \Bigg) \\
&\leq L(1 + \alpha L) \cdot \|\theta_1 - \theta_2\|
\end{aligned}
\tag{36}
$$

By Assumption A.6 and A.7, for $P_2$, we have:

$$P_2 \leq \underbrace{\left\| \nabla_\theta^2 \mathcal{L}_i(\theta_1) \right\|}_{L} \cdot \underbrace{\left\| \nabla_{\theta_1'} \mathcal{L}_i(\theta_1') - \nabla_{\theta_2'} \mathcal{L}_i(\theta_2') \right\|}_{=P_1} + \underbrace{\left\| \nabla_\theta^2 \mathcal{L}_i(\theta_1) - \nabla_\theta^2 \mathcal{L}_i(\theta_2) \right\|}_{\leq \rho \|\theta_1 - \theta_2\|} \cdot \underbrace{\left\| \nabla_{\theta_2'} \mathcal{L}_i(\theta_2') \right\|}_{\leq \mu} \tag{37}$$

$$\leq L \cdot L(1 + \alpha L) \|\theta_1 - \theta_2\| + \rho \|\theta_1 - \theta_2\| \cdot \mu$$

By substituting $P_1$ and $P_2$ back into the Equation 35, we obtain:

$$\|g_i(\theta_1) - g_i(\theta_2)\| \leq \left[ L(1 + \alpha L) + \alpha L^2(1 + \alpha L) + \rho \mu \right] \cdot \|\theta_1 - \theta_2\| \tag{38}$$

Taking the expectation gives:

$$L_F = L(1 + \alpha L) + \alpha \rho \mu \tag{39}$$

$\square$

**Lemma A.11.** *(Smoothness Inequality): If $F$ is $L$-smooth, then for any $x, y \in \mathbb{R}^d$, the following inequality holds:*

$$F(y) \leq F(x) + \langle \nabla_x F(x), y - x \rangle + \frac{L}{2} \|y - x\|^2 \tag{40}$$

*Proof.* We start from the fundamental theorem of calculus by considering the straight-line path between $x$ and $y$. Define the auxiliary function:

$$\psi(t) = F\big(x + t(y - x)\big), \qquad t \in [0, 1] \tag{41}$$

Differentiating $\psi(t)$ with respect to $t$, we obtain:

$$\psi'(t) = \big\langle \nabla_x F\big(x + t(y - x)\big), y - x \big\rangle \tag{42}$$

Thus,

$$F(y) - F(x) = \psi(1) - \psi(0) = \int_0^1 \psi'(t)\, dt = \int_0^1 \big\langle \nabla_x F\big(x + t(y - x)\big), y - x \big\rangle dt \tag{43}$$

By introducing the term $\langle \nabla_x F(x), y - x \rangle$ and applying the distributive property of inner products, we obtain:

$$F(y) - F(x) = \langle \nabla_x F(x), y - x \rangle + \int_0^1 \underbrace{\big\langle \nabla_x F\big(x + t(y - x)\big) - \nabla_x F(x), y - x \big\rangle}_{P_1} dt \tag{44}$$

By applying the Cauchy-Schwarz inequality ($\langle a, b \rangle \leq \|a\| \cdot \|b\|$) to $P_1$, we obtain:

$$\begin{aligned} P_1 &\leq \underbrace{\left\| \nabla_x F\big(x + t(y - x)\big) - \nabla_x F(x) \right\|}_{L-Smooth} \cdot \|y - x\| \\ &\leq L \|x + t(y - x) - x\| \cdot \|y - x\| \\ &= Lt \|y - x\| \cdot \|y - x\| \\ &= Lt \|y - x\|^2 \end{aligned} \tag{45}$$

By substituting $P_1$ back into Equation 44, we obtain:

$$\begin{aligned} F(y) - F(x) &\leq \langle \nabla_x F(x), y - x \rangle + \int_0^1 Lt \|y - x\|^2\, dt \\ &= \langle \nabla_x F(x), y - x \rangle + L \|y - x\|^2 \int_0^1 t\, dt \\ &= \langle \nabla_x F(x), y - x \rangle + \frac{L}{2} \|y - x\|^2 \end{aligned} \tag{46}$$

$\square$

**Lemma A.12.** *(Upper Bound on the Variance of the Stochastic Gradient):* *Under Assumptions A.6-A.8, for any $\theta \in \mathbb{R}^d$ it holds that:*

$$\mathbb{E}\big[\|\widehat{g}(\theta)\|^2\big] \leq 2\|\nabla_\theta F(\theta)\|^2 + \frac{2\sigma^2}{M} \tag{47}$$

*Proof.* Let $\mathbb{E}\big[\|\widehat{g}(\theta)\|^2\big]$ be taken over i.i.d. samples and introduce the term $\|\nabla_\theta F(\theta)\|^2$, we obtain:

$$
\begin{aligned}
\mathbb{E}\big[\|\widehat{g}(\theta)\|^2\big] &= \mathbb{E}\Bigg[\bigg\|\frac{1}{M}\sum_{i=1}^{M} g_i(\theta)\bigg\|^2\Bigg] \\
&= \|\nabla_\theta F(\theta)\|^2 + \frac{1}{M}\cdot\frac{1}{M}\underbrace{\sum_{i=1}^{M}\mathbb{E}\big[\|g_i(\theta) - \nabla_\theta F(\theta)\|^2\big]}_{\leq \sigma^2} \\
&\leq \|\nabla_\theta F(\theta)\|^2 + \frac{\sigma^2}{M} \\
&\leq 2\|\nabla_\theta F(\theta)\|^2 + \frac{2\sigma^2}{M}
\end{aligned}
\tag{48}
$$

$\square$

### A.3.4. THEOREM PROOF

**Theorem A.13.** *(Convergence Rate of HDCL): Based on Assumptions A.6-A.9 and Lemmas A.10-A.12, let $\alpha = \frac{1}{2L_F\sqrt{T}}$. Then there exists a constant $C = 4L_F\big(F(\theta_0) - F^*\big) + \frac{\sigma^2}{M}$ such that:*

$$\frac{1}{T}\sum_{t=0}^{T-1}\mathbb{E}\big[\|\nabla_\theta F(\theta_t)\|^2\big] \leq \frac{C}{\sqrt{T}} \tag{49}$$

*where $F^* = \inf_{\theta \in \mathbb{R}^d} F(\theta)$ represents the global minimum.*

*Proof.* Step 1: Smoothness Expansion. By $L_F$-smoothness (Lemmas A.10 and A.11, and Definition A.5):

$$
\begin{aligned}
F(\theta_{t+1}) &\leq F(\theta_t) + \Big\langle \nabla_\theta F(\theta_t), \underbrace{\theta_{t+1} - \theta_t}_{-\alpha\widehat{g}_t}\Big\rangle + \frac{L_F}{2}\Big\|\underbrace{\theta_{t+1} - \theta_t}_{-\alpha\widehat{g}_t}\Big\|^2 \\
&= F(\theta_t) + \alpha\langle\nabla_\theta F(\theta_t), \widehat{g}_t\rangle + \frac{L_F\alpha^2}{2}\|\widehat{g}_t\|^2
\end{aligned}
\tag{50}
$$

Step 2: Conditional Expectation. Given $\theta_t$, taking the expectation over all $M$ subsets yields:

$$\mathbb{E}_{M|\theta_t}\big[\langle\nabla_\theta F(\theta_t), \widehat{g}_t\rangle\big] = \langle\nabla_\theta F(\theta_t), \nabla_\theta F(\theta_t)\rangle = \|\nabla_\theta F(\theta_t)\|^2 \tag{51}$$

Then, taking the conditional expectation of Equation 50 and applying Lemma A.12, we obtain:

$$
\begin{aligned}
\mathbb{E}_{M|\theta_t}\big[F(\theta_{t+1})\big] &\leq F(\theta_t) - \alpha\|\nabla_\theta F(\theta_t)\|^2 + \frac{L_F\alpha^2}{2}\underbrace{\mathbb{E}_{M|\theta_t}\big[\|\widehat{g}_t\|^2\big]}_{\leq 2\|\nabla_\theta F(\theta_t)\|^2 + \frac{2\sigma^2}{M}} \\
&\leq F(\theta_t) - \alpha\|\nabla_\theta F(\theta_t)\|^2 + L_F\alpha^2\left(\|\nabla_\theta F(\theta_t)\|^2 + \frac{\sigma^2}{M}\right) \\
&= F(\theta_t) - \alpha(1 - L_F\alpha)\|\nabla_\theta F(\theta_t)\|^2 + \frac{L_F\alpha^2\sigma^2}{M}
\end{aligned}
\tag{52}
$$

Step 3: Full Expectation and Summation. Taking the full expectation $\mathbb{E}[\cdot]$, we obtain:

$$\mathbb{E}\Big[\mathbb{E}_{M|\theta_t}\big[F(\theta_{t+1})\big]\Big] \leq \mathbb{E}\Big[F(\theta_t) - \alpha(1 - L_F\alpha)\|\nabla_\theta F(\theta_t)\|^2 + \frac{L_F\alpha^2\sigma^2}{M}\Big] \tag{53}$$

$$\mathbb{E}\big[F(\theta_{t+1})\big] \leq \mathbb{E}\big[F(\theta_t)\big] - \alpha(1 - L_F\alpha)\mathbb{E}\big[\|\nabla_\theta F(\theta_t)\|^2\big] + \frac{L_F\alpha^2\sigma^2}{M}$$

Let $\triangle_t = \mathbb{E}\big[F(\theta_{t+1})\big]$, summing over $t = 0, 1, \cdots, T-1$, we obtain:

$$\triangle_T \leq \triangle_0 - \alpha(1 - L_F\alpha)\sum_{t=0}^{T-1}\mathbb{E}\big[\|\nabla_\theta F(\theta_t)\|^2\big] + \frac{TL_F\alpha^2\sigma^2}{M} \tag{54}$$

Since $F^* \leq \triangle_T$, rearranging terms yields:

$$\alpha(1 - L_F\alpha)\sum_{t=0}^{T-1}\mathbb{E}\big[\|\nabla_\theta F(\theta_t)\|^2\big] \leq F(\theta_0) - F^* + \frac{TL_F\alpha^2\sigma^2}{M} \tag{55}$$

Step 4: Learning rate selection and convergence rate. Let $\alpha = \frac{1}{2L_F\sqrt{T}}$. When $T \geq 1$, we have:

$$1 - L_F\alpha \geq 1 - \frac{1}{2\sqrt{T}} \geq \frac{1}{2} \tag{56}$$

Substituting this into Equation 55, we obtain:

$$\frac{1}{2L_F\sqrt{T}} \cdot \frac{1}{2} \cdot \sum_{t=0}^{T-1}\mathbb{E}\big[\|\nabla_\theta F(\theta_t)\|^2\big] \leq F(\theta_0) - F^* + \frac{TL_F\sigma^2}{M} \cdot \frac{1}{4L_F^2 T}$$

$$\frac{4L_F\sqrt{T}}{T} \cdot \Big(\frac{1}{2L_F\sqrt{T}} \cdot \frac{1}{2} \cdot \sum_{t=0}^{T-1}\mathbb{E}\big[\|\nabla_\theta F(\theta_t)\|^2\big]\Big) \leq \Big(F(\theta_0) - F^* + \frac{TL_F\sigma^2}{M} \cdot \frac{1}{4L_F^2 T}\Big) \cdot \frac{4L_F\sqrt{T}}{T}$$

$$\frac{1}{T} \cdot \sum_{t=0}^{T-1}\mathbb{E}\big[\|\nabla_\theta F(\theta_t)\|^2\big] \leq \frac{4L_F\big(F(\theta_0) - F^*\big)}{\sqrt{T}} + \frac{\sigma^2}{M\sqrt{T}} \tag{57}$$

$$\frac{1}{T} \cdot \sum_{t=0}^{T-1}\mathbb{E}\big[\|\nabla_\theta F(\theta_t)\|^2\big] \leq \Big(\underbrace{4L_F\big(F(\theta_0) - F^*\big) + \frac{\sigma^2}{M}}_{C}\Big) \cdot \frac{1}{\sqrt{T}}$$

$\square$

Theorem A.13 indicates that, with an appropriately chosen learning rate, the HDCL algorithm guarantees convergence of the model parameters to a point that generalizes well across the heterogeneous subsets.

## B. Algorithm Analysis

### B.1. Optimal M Search Algorithm

The pseudocode for searching the optimal value of $M$ is provided in Algorithm 1, where $\epsilon$ denotes the BIC score of each subset, and quantities marked with the superscript $*$ indicate their optimal values.

#### B.1.1. COMPLEXITY

The GMM is optimized using the Expectation-Maximization (EM) algorithm. Given a representation set of size $n$, representation dimension $d$, and a specified number of clusters $k$, the computational cost of GMM clustering is $O(nkd)$ when Gaussian densities are computed using diagonal covariance matrices. In Algorithm 1, let the representation set size be $N$, feature dimension $D$, the initial search interval be $[s, e]$, and the step size be $\tau$. In each iteration of the while-loop, $\frac{e-s}{\tau}$ clustering operations are performed (Line 2). The average number of clusters evaluated within the interval $[s, e]$ is approximately $\frac{s+e}{2}$. Therefore, the total clustering cost of a single while-loop is $O(\frac{e-s}{\tau} \cdot N \cdot \frac{s+e}{2} \cdot D) < O(\frac{e^2}{2\tau}ND)$. After each while-loop, both the search interval and the step size are halved (Line 13), implying that the number of while-loop iterations is $\log\tau$. Combining the above results, the overall computational complexity of Algorithm 1 is $O(\frac{e^2}{2\tau}ND \cdot \log\tau)$.

**Algorithm 1** FindM

**Input**: $\{r_i\}_{i=1}^N, [s, e], \tau$
**Output**: $M$

1:   $M^* \leftarrow 0$
2:   **while** $\tau > 0$ **and** $s < e$ **do**
3:     $\epsilon^* \leftarrow \infty$
4:     $m \leftarrow s$
5:     **while** $m \leq e$ **do**
6:       $\epsilon \leftarrow GMM(m, \{r_i\}_{i=1}^N)$
7:       **if** $\epsilon < \epsilon^*$ **then**
8:         $\epsilon^* \leftarrow \epsilon$
9:         $M^* \leftarrow m$
10:      **end if**
11:      $m \leftarrow m + \tau$
12:     **end while**
13:     $s, e, \tau \leftarrow \frac{s+M^*}{2}, \frac{e+M^*}{2}, \frac{\tau}{2}$
14: **end while**
15: **return** $M^*$

**Algorithm 2** HDCL

**Input**: $\{c_i^s\}_{i=1}^M, \{c_i^q\}_{i=1}^M, \{\phi_i(\cdot; \theta_i)\}_{i=1}^M, \phi_g(\cdot; \theta_g), \theta_{\text{SWA}}$
**Output**: $\phi_g(\theta_g)$

1:   **for** $t \leftarrow 1$ **to** $N_g$ **do**
2:     $\{\theta_i\}_{i=1}^M \leftarrow \theta_g$
3:     **for** $i \leftarrow 1$ **to** $M$ **do**
4:       **for** $k \leftarrow 1$ **to** $N_p$ **do**
5:         $\mathcal{L}_i^s \leftarrow \phi_i(c_i^s; \theta_i)$
6:         $\theta_i \leftarrow \theta_g - \alpha \nabla_{\theta_g} \mathcal{L}_i^s$
7:       **end for**
8:     **end for**
9:     $\mathcal{L}_{HDCL} \leftarrow \{\mathcal{L}_i^q \leftarrow \phi_i(c_i^q; \theta_i)\}_{i=1}^M$
10:    **for** $\mathcal{L}_i^q$ **in** $\mathcal{L}_{HDCL}$ **do**
11:      $\theta_g \leftarrow \theta_g - \alpha \omega_i \nabla_{\theta_g} \mathcal{L}_i^q$
12:    **end for**
13:    $\theta_{\text{SWA}}^{t+1} \leftarrow \frac{\theta_{\text{SWA}}^t \cdot t + \theta_g}{t+1}$
14: **end for**
15: **return** $\phi_g$

## B.2. HDCL Algorithm

The HDCL aims to jointly learn across heterogeneous subsets and ultimately obtain a model with strong generalization capability over all subsets. The pseudocode is provided in Algorithm 2, where $N_g$ and $N_p$ denote the number of global training epochs and the number of proxy model training epochs, respectively.

### B.2.1. COMPLEXITY

Each training iteration of the HDCL consists of two stages: training $M$ proxy models and updating the prediction model. Both the proxy models and the prediction model share the same MLP architecture. Suppose the MLP has $L$ layers, with the width of the $l$-th layer denoted as $H_l$(where the input layer width $H_0$ equals the dimensionality of the representations). Then, the complexity for a single forward pass, backward pass, and parameter update is $O\left(\sum_{l=1}^L H_{l-1}H_l\right)$. In Algorithm 2, training the proxy models involves performing $N_g \cdot N_p \cdot M$ forward and backward passes along with parameter updates (Lines 5-6), resulting in an overall complexity of $O\left(N_g N_p M \sum_{l=1}^L H_{l-1}H_l\right)$ for the proxy training stage. Subsequently, updating the prediction model requires $N_g \cdot M$ parameter updates (Line 11), with a complexity of $O\left(N_g M \sum_{l=1}^L H_{l-1}H_l\right)$. Therefore, the total complexity of Algorithm 2 is dominated by the proxy training stage and can be expressed as $O\left(N_g N_p M \sum_{l=1}^L H_{l-1}H_l\right)$.

## C. Implementation Details

*MixEmo* is implemented based on PyTorch. All experiments are conducted on a server running Ubuntu 20.04, equipped with an AMD EPYC 7543 CPU and an NVIDIA A100 GPU. The hyperparameter settings of *MixEmo* are summarized in Table 4.

*Table 4.* Configurations of *MixEmo*

| Dataset | Encoder | Backbone (Epochs/$\alpha$) | HDCL (Epochs/$\alpha$) | Batch | $d$ | $M$ | $m/n$ |
|---------|---------|------------|------------|-------|-----|-----|-------|
| DEAP | CNN | $100/1e-4$ | $10\text{-}1000/1e-4$ | 256 | 1024 | 5 | $M-1$ |
| SEED | CNN | $100/1e-4$ | $10\text{-}1000/1e-4$ | 256 | 1024 | 5 | $M-1$ |
| WESAD | MLP | $10/1e-3$ | $10\text{-}1000/1e-4$ | 512 | 1024 | 5 | $M-1$ |
| SE-R1510 | LSTM | $1500/1e-3$ | $20\text{-}1500/1e-4$ | 128 | 512 | 5 | $M-1$ |
| SE-F1530 | LSTM | $1500/1e-3$ | $20\text{-}1500/1e-4$ | 128 | 512 | 5 | $M-1$ |

## C.1. Backbone

The backbone model adopts an encoder–classifier architecture and is optimized using Adam. Specifically, the type of encoder and its hyperparameter settings are listed in Table 4, where $d$ denotes the representation dimension. For all datasets, the classifier is implemented as an MLP, with its network structure detailed in Table 5.



*Table 5.* Classifier architecture for each dataset

| Dataset | Layer | Input | Output |
|---------|-------|-------|--------|
| DEAP | Linear | $d$ | $Y$ |
| SEED | Linear | $d$ | $Y$ |
| WESAD | Linear | $d$ | $Y$ |
| SE-R1510 | Linear | $d$ | $Y$ |
| SE-F1530 | Linear | $d$ | $Y$ |

*Table 6.* Architecture of proxy models and prediction model

| Layer | Input | Output | Activate |
|-------|-------|--------|----------|
| Linear | $d$ | $2d$ | ReLU |
| Linear | $2d$ | $4d$ | ReLU |
| Linear | $4d$ | $4d$ | ReLU |
| Linear | $4d$ | $2d$ | ReLU |
| Linear | $2d$ | $Y$ | - |



## C.2. UDG

The detailed configurations adopted for each dataset are summarized in Table 4. When constructing synthetic unseen distributions, it is necessary to ensure that each synthesized subset contains a sufficient number of training samples for support-query partitioning while satisfying the constraint condition ($m/n \geq M - 1$). Based on this consideration, we adopt the following two measures in implementation to mitigate its potential impact: 1) The sampling ratio between the primary subset and the auxiliary subset ($m/n$) is uniformly set to $M - 1$, where $m$ denotes the number of samples in the primary subset; 2) The ratio between the support set and the query set is fixed at 4:1 to ensure that the support set contains sufficient information, thereby alleviating the adverse effects caused by distributional shift.

## C.3. HDCL

The HDCL optimizes the proxy models and the prediction model through an alternating training scheme, using SGD as the optimizer, with the number of iterations specified in Table 4. To facilitate efficient weight updates during training, all proxy models and the prediction model share the same MLP architecture, with detailed configurations provided in Table 6.

# D. Dataset Details

To demonstrate the weakly modality-dependent nature of *MixEmo*, we conducted experiments on five emotion datasets covering three different modalities. Detailed information about each dataset is provided as follows:

- **DEAP** (Koelstra et al., 2012): This dataset records EEG signals from 32 participants while they watched 40 one-minute-long music video clips. Participants rated each video based on arousal (1-9), valence (1-9), like/dislike, dominance, and familiarity. For this experiment, the arousal and valence ratings were selected as the primary indicators, binarized using a threshold of 5, resulting in a dataset with four distinct emotion labels. The dataset is available for download at: https://www.eecs.qmul.ac.uk/mmv/datasets/deap/.

- **SEED** (Zheng & Lu, 2015): This dataset captures EEG signals from 15 participants while they watched four-minute-long movie clips. After viewing each video, participants were required to evaluate their emotions as either positive, neutral, or negative. The dataset is available for download at: https://bcmi.sjtu.edu.cn/~seed/seed.html.

- **WESAD** (Schmidt et al., 2018): This dataset contains physiological and motion data collected from 15 participants using wrist-worn and chest-worn devices. The recorded modalities include blood volume pulse, electrocardiogram, electrodermal activity, electromyogram, respiration, body temperature, and three-axis acceleration. The dataset categorizes emotional states into three types: neutral, stressed, and amused. These emotional states were derived using several established questionnaires. The dataset is available for download at: https://ubi29.informatik.uni-siegen.de/usi/data_wesad.html.

- **SE-R1510** (Chen, 2023): This dataset contains spatiotemporal trajectories of 15 participants recorded over 10 to 12 consecutive days. During the data collection period (6:00-24:00 daily), participants performed at least three

self-reported emotion evaluations each day. Throughout this period, all participants' events were restricted to within a school campus to investigate the impact of limited mobility on emotions.

- **SE-F1530** (Chen, 2023): This dataset includes spatiotemporal trajectories of 15 participants recorded over 29 to 30 consecutive days. During the data collection period (6:00-24:00 daily), participants performed at least three self-reported emotion evaluations each day. In contrast to SE-R1510, participants' events were unrestricted, allowing for the study of the effects of diverse societal events under normal conditions on emotional states.

## E. Baseline Details

The state-of-the-art methods we selected include both data augmentation approaches and cross-subject emotion recognition methods specifically designed for EEG signals. A detailed description of these methods is provided below:

- **Mixup** (Zhang et al., 2017): A simple yet widely adopted data augmentation technique whose core idea is to construct virtual training samples by forming convex combinations of paired samples and their corresponding labels during training. This strategy encourages neural networks to learn smoother and approximately linear decision boundaries between training examples, thereby improving generalization performance. The source code is available at: https://github.com/facebookresearch/mixup-cifar10.

- **MAML** (Finn et al., 2017): A classic meta-learning paradigm that is compatible with any model optimized via gradient-based methods and is applicable to a wide range of learning tasks, including classification, regression, and reinforcement learning. Its objective is to meta-train a model across multiple tasks so as to learn a highly transferable parameter initialization, enabling rapid adaptation to new tasks with only a few training samples and a limited number of gradient update steps. The source code is available at: https://github.com/cbfinn/maml.

- **CTGAN** (Xu et al., 2019): A GAN designed for generating tabular data, addressing challenges such as continuous columns with diverse patterns and imbalanced discrete columns. CTGAN introduces a mode-specific normalization technique that converts continuous values with arbitrary ranges and distributions into bounded vector representations suitable for neural networks. Additionally, it employs conditional generators and sampling-based training to handle class imbalance in the training data. The source code is available at: https://github.com/sdv-dev/CTGAN.

- **GT-GAN** (Jeon et al., 2022): A GAN framework for generating time series, incorporating techniques from neural controlled differential equations, continuous-time stochastic processes, and more. GT-GAN can synthesize various types of time series without requiring changes to the model architecture or parameters. The source code is available at: https://github.com/Jinsung-Jeon/GT-GAN.

- **PCF-GAN** (Lou et al., 2023): A GAN for time series generation that introduces an efficient discriminator to differentiate time series distributions. The authors establish the theoretical foundation of the Path Characteristic Function, demonstrating its properties such as characteristicness, boundedness, differentiability with respect to generator parameters, and weak continuity. PCF-GAN incorporates the PCF as a principled representation of time series distributions into the discriminator, enhancing generation performance. Additionally, it features efficient initialization and optimization strategies to improve discriminative power and accelerate training. The source code is available at: https://github.com/DeepIntoStreams/PCF-GAN.

- **InfoTS** (Luo et al., 2023): A model for time series generation designed to address the structural diversity and complexity of time series, which are often difficult to identify manually. Based on information theory, the authors propose a feasible data augmentation standard and introduce a information-aware contrastive learning approach. This method adaptively selects the optimal augmentation strategy for time series representation learning. The source code is available at: https://github.com/chengw07/InfoTS.

- **TimeDP** (Huang et al., 2025): A diffusion model-based method for multi-domain augmentation. TimeDP defines prototype vectors for each domain, where each prototype represents the fundamental temporal characteristics of that domain. During the sampling process, TimeDP extracts a small number of samples from the target domain to generate domain-specific prompts, which are then used as conditions for generating time series samples. The source code is available at: https://github.com/microsoft/TimeCraft/tree/main/.

- **EEGMatch** (Zhou et al., 2025): A semi-supervised transfer learning framework designed to leverage both labeled and unlabeled EEG data. It first introduces an EEG confusion-based data augmentation method to generate more diverse samples. Next, it employs a paired learning strategy to bridge the gap between prototype-level and instance-level pairwise learning. Finally, a semi-supervised multi-domain generalization mechanism is incorporated to align data representations across domains by mitigating distribution mismatches. The source code is available at: `https://github.com/KAZABANA/EEGMatch`.

- **DMMR** (Wang et al., 2024): A denoising mixed mutual reconstruction model that uses a two-stage pretraining and fine-tuning strategy. In the pretraining stage, self-supervised learning via an autoencoder guides the encoder to extract subject-invariant features. In the fine-tuning stage, an emotion classifier is integrated to capture emotion-relevant features. Additionally, a hidden-layer mixed data augmentation method is introduced to enhance noise robustness and alleviate source data scarcity. The source code is available at: `https://github.com/CodeBreathing/DMMR`.

- **MoGE** (Liu et al., 2024c): A sparse Mixture-of-Graph-Experts model that explores domain generalization from a neural network architecture perspective. A router assigns each EEG channel to a specific expert, decomposing complex brain signals into relatively independent functional regions, which facilitates the extraction of region-specific features. The source code is available at: `https://github.com/XuanhaoLiu/MoGE`.

- **EmT** (Ding et al., 2025): A framework designed to capture key long-term contextual information associated with emotion cognition processes. Specifically, it first converts EEG signals into temporal graph structures via a temporal graph construction module, thereby generating a sequence of EEG feature maps. Subsequently, an innovative residual multi-view pyramid graph convolutional network is proposed to learn dynamic graph representations of EEG feature maps within the sequence, which are then further integrated into a unified global token for emotion modeling. The source code is available at: `https://github.com/yi-ding-cs/EmT`.

To ensure a fair comparison, the inputs are uniformly controlled according to the applicability of different methods: 1) When comparing with general data augmentation methods, all approaches are applied in the representation space, where encoded sample representations are used as inputs; 2) When comparing with EEG-specific methods as well as Mixup and MAML, all approaches are trained and evaluated directly on raw samples.

## F. Detailed Experimental Results

### F.1. Performance of the Backbone

To demonstrate that the selected backbone possesses strong inherent representation capabilities, we report its performance under a non-cross-subject setting (😄). Furthermore, to illustrate the performance improvement brought by *MixEmo* in cross-subject scenarios, we also provide the backbone's results under a cross-subject setting (🥴). The complete results are presented in Table 7.

### F.2. Superiority Experiment

The complete results of the superiority experiments are presented in Tables 9 and 10, where the **bold** and underlined values denote the best and second-best performance, respectively.

### F.3. Ablation Experiment

To quantify the individual contributions of each component within *MixEmo*, we conducted ablation studies on the key components of the framework. The complete results are reported in Tables 11 and 13.

### F.4. Sensitivity Experiment

We analyze the sensitivity of *MixEmo* to the number of subsets $M$, the representation dimension $d$, and the sampling ratio $m/n$. The complete experimental results are reported in Tables 8, 12, and 14.

*Table 7.* Performance of backbones under non-cross-subject (😊) and cross-subject (😵) settings (%)

| Dataset | Setting 😊 | Setting 😵 | Accuracy | Precision | Recall | F1 |
|---|---|---|---|---|---|---|
| DEAP | ✓ | | 88.25 | 88.30 | 88.26 | 88.27 |
| | | ✓ | 25.15 | 25.03 | 24.45 | 25.03 |
| SEED | ✓ | | 68.49 | 68.49 | 68.49 | 68.49 |
| | | ✓ | 44.26 | 47.23 | 44.26 | 44.54 |
| WESAD | ✓ | | 97.01 | 97.02 | 97.01 | 97.02 |
| | | ✓ | 52.34 | 51.15 | 52.34 | 50.31 |
| SE-R1510 | ✓ | | 69.70 | 66.47 | 68.01 | 63.05 |
| | | ✓ | 48.91 | 33.78 | 44.04 | 27.34 |
| SE-F1530 | ✓ | | 60.67 | 55.91 | 54.99 | 55.32 |
| | | ✓ | 30.23 | 35.70 | 33.79 | 27.87 |

*Table 8.* Sensitivity analysis with respect to the number of subsets $M$ (%)

| Dataset | $M$ | Accuracy | Precision | Recall | F1 |
|---|---|---|---|---|---|
| DEAP | 3 | 24.75 | 24.68 | 24.75 | 24.54 |
| | 4 | 25.21 | 24.96 | 25.21 | 24.80 |
| | 5 | **29.52** | **31.56** | **29.53** | **30.34** |
| | 6 | 26.25 | 26.10 | 26.25 | 25.83 |
| | 7 | 26.58 | 26.10 | 26.60 | 25.22 |
| SEED | 3 | 50.93 | 51.91 | 50.93 | 48.94 |
| | 4 | 54.44 | 55.51 | 54.44 | 53.17 |
| | 5 | **62.04** | **63.23** | **62.04** | **60.32** |
| | 6 | 59.44 | 60.20 | 59.44 | 58.61 |
| | 7 | 59.07 | 59.61 | 59.07 | 58.38 |
| WESAD | 15 | 63.74 | 63.00 | 63.74 | 63.12 |
| | 16 | 66.89 | 71.94 | 66.89 | 65.60 |
| | 17 | **70.37** | 70.74 | **70.37** | 67.58 |
| | 18 | 67.22 | **72.01** | 67.22 | 66.09 |
| | 19 | 68.07 | 66.82 | 68.07 | 65.71 |
| SE-R1510 | 3 | 54.74 | 42.29 | 39.67 | 40.07 |
| | 4 | 56.96 | 41.81 | 38.74 | 39.72 |
| | 5 | **62.50** | **51.29** | 52.98 | **48.74** |
| | 6 | **62.50** | 50.15 | **53.07** | 48.68 |
| | 7 | 61.76 | 47.78 | 42.06 | 44.14 |
| SE-F1530 | 3 | 46.95 | 43.04 | 42.75 | 41.84 |
| | 4 | 49.62 | 44.65 | 46.68 | 44.22 |
| | 5 | **52.67** | 45.80 | 47.09 | **45.80** |
| | 6 | 51.91 | **45.86** | **47.24** | 45.68 |
| | 7 | 51.53 | 44.61 | 45.16 | 44.40 |

*Table 9.* Complete comparison between *MixEmo* and state-of-the-art general methods (%)

| Dataset | Method | Accuracy | Precision | Recall | F1 |
|---|---|---|---|---|---|
| DEAP | Mixup | 27.32 | 24.24 | 24.39 | 23.44 |
| | MAML | 25.04 | 24.86 | 25.04 | 24.73 |
| | CTGAN | 26.08 | 25.79 | 26.08 | 25.25 |
| | GT-GAN | 25.25 | 25.20 | 25.49 | 24.44 |
| | PCF-GAN | 26.29 | 25.92 | 26.29 | 25.55 |
| | InfoTS | 25.58 | 25.98 | 25.53 | 25.32 |
| | TimeDP | 25.20 | 24.97 | 25.29 | 23.99 |
| | *MixEmo* | **29.52** | **31.56** | **29.53** | **30.34** |
| SEED | Mixup | 51.20 | 50.74 | 50.96 | 50.82 |
| | MAML | 47.22 | 53.20 | 47.22 | 46.63 |
| | CTGAN | 58.52 | 59.84 | 58.52 | 57.40 |
| | GT-GAN | 57.78 | 58.84 | 57.78 | 56.07 |
| | PCF-GAN | 55.19 | 55.46 | 55.19 | 53.76 |
| | InfoTS | 47.04 | 51.81 | 47.04 | 47.08 |
| | TimeDP | 48.33 | 51.38 | 48.33 | 47.71 |
| | *MixEmo* | **62.04** | **63.23** | **62.04** | **60.32** |
| WESAD | Mixup | 43.17 | 41.47 | 43.17 | 41.80 |
| | MAML | 48.41 | 52.69 | 48.41 | 40.56 |
| | CTGAN | 66.59 | 65.21 | 66.59 | 64.52 |
| | GT-GAN | 67.59 | 67.14 | 67.59 | 65.66 |
| | PCF-GAN | 61.56 | 61.95 | 61.56 | 61.47 |
| | InfoTS | 47.26 | 47.91 | 47.26 | 47.54 |
| | TimeDP | 55.89 | 54.23 | 55.89 | 54.34 |
| | *MixEmo* | **70.37** | **70.74** | **70.37** | **67.58** |
| SE-R1510 | Mixup | 57.55 | 46.53 | 46.16 | 44.13 |
| | MAML | 44.57 | 34.67 | 42.41 | 25.56 |
| | CTGAN | 58.95 | 47.98 | 48.10 | 47.95 |
| | GT-GAN | 55.88 | **53.04** | **54.42** | 44.54 |
| | PCF-GAN | 57.30 | 40.35 | 45.28 | 38.52 |
| | InfoTS | 53.68 | 29.50 | 35.94 | 32.41 |
| | TimeDP | 54.21 | 29.60 | 30.21 | 28.21 |
| | *MixEmo* | **62.50** | 50.15 | 53.07 | **48.68** |
| SE-F1530 | Mixup | 46.63 | 40.02 | 39.00 | 39.06 |
| | MAML | 48.09 | 43.08 | 43.83 | 42.42 |
| | CTGAN | 49.62 | 45.26 | **47.93** | 44.70 |
| | GT-GAN | 47.88 | 41.87 | 42.10 | 41.14 |
| | PCF-GAN | 44.44 | 42.87 | 44.05 | 39.60 |
| | InfoTS | 43.37 | 39.67 | 40.22 | 39.69 |
| | TimeDP | 39.31 | 35.50 | 38.10 | 33.47 |
| | *MixEmo* | **52.67** | **45.80** | 47.09 | **45.80** |

# G. Related Work

## G.1. Based on data augmentation

Data augmentation methods are widely regarded as effective means for improving model generalization and constitute one of the mainstream research directions for addressing cross-subject emotion recognition. Regarding EEG data augmentation, (Zhang et al., 2024b) proposed an EEG sample generation method based on GANs and self-supervised learning to address sample imbalance issues. For speech data augmentation, (Wang et al., 2020) developed a GAN and Variational Autoencoder framework conditioned on different input vectors. (Su & Lee, 2023) proposed a corpus-aware emotional cyclic wide-area model, which generates semantically rich target data in an unsupervised cross-corpus manner and aggregates source data information using a corpus-aware attention mechanism. For addressing emotion category imbalance in textual data, (Meng et al., 2024) combined multimodal approaches to develop a cross-modal sample generation method based on GANs. Overall, existing work typically assumes that datasets exhibit distribution consistency and enhances model generalization through GAN-based data augmentation methods. However, datasets often contain multiple subsets with significant distribution differences, which limits the performance of such methods.

*Table 10.* Complete comparison between *MixEmo* and state-of-the-art EEG-specific cross-subject emotion recognition methods (%)

| Dataset | DEAP | | | | SEED | | | |
| Method | Accuracy | Precision | Recall | F1 | Accuracy | Precision | Recall | F1 |
|---|---|---|---|---|---|---|---|---|
| EEGMatch | 27.63 | 27.61 | 27.72 | 26.47 | 60.56 | 61.37 | 60.56 | 58.98 |
| DMMR | 27.42 | 27.47 | 27.42 | 27.13 | **62.41** | **64.02** | **62.41** | **61.35** |
| MoGE | 27.28 | 27.22 | 27.43 | 26.23 | 61.30 | 62.55 | 61.30 | 60.22 |
| EmT | 26.83 | 26.78 | 26.83 | 26.63 | 59.26 | 59.23 | 59.26 | 57.87 |
| *MixEmo* | **29.52** | **31.56** | **29.53** | **30.34** | 62.04 | 63.23 | 62.04 | 60.32 |

## G.2. Based on specific emotional modalities

Due to the significant individual differences in physiological signals, there has been a growing body of work on personalized emotion recognition based on such signals in recent years. For example, (Zhao et al., 2021) divided EEG representations into specific and shared components using different encoders and proposed a plug-and-play domain adaptation approach to address inter-subject variability. (Liu et al., 2021) introduced a dynamic differential entropy algorithm to extract features from EEG signals and developed a subject-independent emotion recognition algorithm based on these features. To quantify inter-subject uncertainty, (Song et al., 2023) proposed a variational instance-adaptive graph method that simultaneously captures individual dependencies between different EEG electrodes and estimates latent uncertain information. (Wang et al., 2024) developed a denoising hybrid mutual reconstruction model, utilizing a two-stage pretraining and fine-tuning process to separately learn subject-independent and subject-dependent features. In the context of domain adaptation, (Li et al., 2024b) introduced an unsupervised sub-region alignment adaptive approach. Gu et al. (Gu et al., 2024) proposed a local–global collaborative learning approach based on EEG signals. Their method decomposes the model into two subnetworks, where local subdomain alignment is used to mitigate emotion discrepancies across subjects, and global domain alignment achieves cross-subject feature alignment by minimizing marginal distribution discrepancies. Luo et al. (Luo et al., 2025) introduced a transfer learning framework that dynamically aligns the source and target domains by mapping data into an optimal Grassmann manifold space. Zhang et al. (Zhang et al., 2024a) proposed a meta-transfer learning method that integrates a dual-attention network with K-means clustering, in which the dual-attention network extracts EEG features via channel attention and temporal attention modules, while the meta-transfer learning strategy enables the model to simultaneously capture shared patterns across subjects and subject-specific variations. However, most of these approaches are specifically designed for particular emotion modalities–almost exclusively EEG signals–and therefore exhibit strong modality dependence in both model architecture and feature representation. As a result, they are difficult to generalize or transfer to other emotion modalities, a limitation that becomes increasingly pronounced with the rapid development of multimodal emotion recognition.

*Table 11.* Ablation study of the UDG and HDCL (%)

| Dataset | Component | | Metrics | | | |
|---|---|---|---|---|---|---|
| | UDG | HDCL | Accuracy | Precision | Recall | F1 |
| DEAP | | ✓ | 25.78 | 25.78 | 25.81 | 25.48 |
| | ✓ | | 24.09 | 23.99 | 24.10 | 23.44 |
| | ✓ | ✓ | **29.52** | **31.56** | **29.53** | **30.34** |
| SEED | | ✓ | 49.93 | 49.99 | 49.93 | 49.96 |
| | ✓ | | 50.93 | 50.08 | 50.93 | 50.07 |
| | ✓ | ✓ | **62.04** | **63.23** | **62.04** | **60.32** |
| WESAD | | ✓ | 50.53 | 50.29 | 50.53 | 50.09 |
| | ✓ | | 53.78 | 53.00 | 53.78 | 53.37 |
| | ✓ | ✓ | **70.37** | **70.74** | **70.37** | **67.58** |
| SE-R1510 | | ✓ | 46.73 | 28.11 | 27.97 | 27.86 |
| | ✓ | | 59.80 | 42.11 | 40.94 | 39.51 |
| | ✓ | ✓ | **62.50** | **50.15** | **53.07** | **48.68** |
| SE-F1530 | | ✓ | 34.60 | 34.69 | 34.84 | 33.27 |
| | ✓ | | 38.46 | 36.04 | 36.04 | 30.06 |
| | ✓ | ✓ | **52.67** | **45.80** | **47.09** | **45.80** |

*Table 13.* Ablation study of the WGU and SWA (%)

| Dataset | Component | | Metrics | | | |
|---|---|---|---|---|---|---|
| | WGU | SWA | Accuracy | Precision | Recall | F1 |
| DEAP | | ✓ | 27.08 | 26.80 | 27.08 | 26.34 |
| | ✓ | | 23.92 | 23.72 | 23.92 | 23.57 |
| | ✓ | ✓ | **29.52** | **31.56** | **29.53** | **30.34** |
| SEED | | ✓ | 58.15 | 58.89 | 58.15 | 57.08 |
| | ✓ | | 48.52 | 47.57 | 48.52 | 45.80 |
| | ✓ | ✓ | **62.04** | **63.23** | **62.04** | **60.32** |
| WESAD | | ✓ | 68.52 | 67.96 | 68.52 | 66.20 |
| | ✓ | | 57.41 | 58.20 | 57.41 | 57.52 |
| | ✓ | ✓ | **70.37** | **70.74** | **70.37** | **67.58** |
| SE-R1510 | | ✓ | 61.46 | 48.69 | 52.26 | 47.92 |
| | ✓ | | 52.63 | 38.62 | 37.60 | 37.86 |
| | ✓ | ✓ | **62.50** | **50.15** | **53.07** | **48.68** |
| SE-F1530 | | ✓ | 51.53 | 44.98 | 45.74 | 44.73 |
| | ✓ | | 40.93 | 27.49 | 28.77 | 25.73 |
| | ✓ | ✓ | **52.67** | **45.80** | **47.09** | **45.80** |

*Table 12.* Sensitivity analysis with respect to the representation dimension $d$ (%)

| Dataset | $d$ | Metrics | | | |
|---|---|---|---|---|---|
| | | Accuracy | Precision | Recall | F1 |
| DEAP | 256 | 26.54 | 26.53 | 26.54 | 26.24 |
| | 512 | 26.71 | 26.44 | 26.73 | 25.44 |
| | 1024 | **29.52** | **31.56** | **29.53** | **30.34** |
| | 2048 | 27.06 | 27.11 | 27.11 | 25.98 |
| SEED | 256 | 58.70 | 59.32 | 58.70 | 56.82 |
| | 512 | 57.59 | 59.04 | 57.59 | 56.67 |
| | 1024 | **62.04** | **63.23** | **62.04** | **60.32** |
| | 2048 | 57.22 | 58.11 | 57.22 | 55.58 |
| WESAD | 256 | 67.63 | 72.35 | 67.63 | 66.52 |
| | 512 | 68.15 | 67.58 | 68.15 | 66.13 |
| | 1024 | **70.37** | 70.74 | **70.37** | **67.58** |
| | 2048 | 67.85 | **72.52** | 67.85 | 66.86 |
| SE-R1510 | 256 | 61.05 | 49.71 | 48.16 | 48.81 |
| | 512 | **62.50** | **51.29** | **52.98** | 48.74 |
| | 1024 | 62.11 | 50.86 | 49.74 | **50.22** |
| | 2048 | 60.87 | 32.57 | 38.21 | 30.15 |
| SE-F1530 | 256 | 51.15 | 44.62 | 45.39 | 44.33 |
| | 512 | **52.67** | 45.80 | 47.09 | 45.80 |
| | 1024 | 50.99 | **52.31** | **54.07** | **50.72** |
| | 2048 | 50.76 | 44.68 | 45.32 | 44.36 |

*Table 14.* Sensitivity analysis with respect to the sampling ratio $m/n$ (%)

| Dataset | $m/n$ | Metrics | | | |
|---|---|---|---|---|---|
| | | Accuracy | Precision | Recall | F1 |
| DEAP | $M-1$ | **29.52** | **31.56** | **29.53** | **30.34** |
| | $M$ | 25.42 | 25.27 | 25.42 | 24.77 |
| | $M+1$ | 24.21 | 23.73 | 24.21 | 23.69 |
| | $M+2$ | 22.79 | 22.58 | 22.79 | 22.53 |
| SEED | $M-1$ | **62.04** | **63.23** | **62.04** | **60.32** |
| | $M$ | 57.96 | 58.53 | 57.96 | 57.05 |
| | $M+1$ | 49.07 | 48.45 | 40.07 | 48.34 |
| | $M+2$ | 41.67 | 48.37 | 41.67 | 40.61 |
| WESAD | $M-1$ | **70.37** | **70.74** | **70.37** | **67.58** |
| | $M$ | 59.85 | 61.47 | 59.85 | 60.33 |
| | $M+1$ | 60.26 | 62.03 | 60.26 | 60.74 |
| | $M+2$ | 53.85 | 54.46 | 53.85 | 52.25 |
| SE-R1510 | $M-1$ | **62.50** | **51.29** | **52.98** | **48.74** |
| | $M$ | 55.79 | 41.94 | 40.22 | 40.67 |
| | $M+1$ | 51.58 | 39.65 | 40.05 | 39.75 |
| | $M+2$ | 47.37 | 27.49 | 34.68 | 29.08 |
| SE-F1530 | $M-1$ | **52.67** | **45.80** | **47.09** | **45.80** |
| | $M$ | 48.85 | 42.46 | 42.91 | 41.95 |
| | $M+1$ | 48.47 | 44.14 | 46.89 | 43.37 |
| | $M+2$ | 40.54 | 26.97 | 28.53 | 25.54 |

