# OpenReview forum: "Enhancing Cross-subject Emotion Recognition via Heterogeneous Distribution Augmentation and Collaborative Learning"
_ICML.cc/2026/Conference — ICML 2026 regular_

### Official Review · Reviewer_MiK3 · 2026-03-09

**Soundness:** 3
**Presentation:** 4
**Significance:** 3
**Originality:** 3
**Overall Recommendation:** 4
**Confidence:** 2

**Summary:**

In this paper, authors resent a training approach to solve the cross-subject emotion recognition, or more precisely, to improve the generalizability of the model to different distributions and modalities. Their approach is interesting s it comprises in 4 phases : extracting the representations from multiple datasets, generating new distribution of these dataset by isolating subjects/modalities, creating support/query balanced versions to train specific models for each, fusing the representations of these specific ones with the initial model. They test it on 5 datasets of emotion recognition. They achieve better performance across datasets, but did not evaluate it on macro f1.

**Compliance With Llm Reviewing Policy:**

Affirmed.

**Final Justification:**

Interesting paper but with missing details. Even though I made a mistake on the related work part, which I aknowledge, it does not change this fact.

**Key Questions For Authors:**

- the support / query sets are balanced according to figure 3. This is a bit surprising as this suppress the $\mathcal{G}_M$ insertion. Is it really balanced or did I misundertstood ? Why doing so if it is?
- in sensitivity analysis, can we derive $d$ from the number of datasets and subjects and modalities ?
- what are the settinigs from t-sne ? diff parameters can results in very different results, did you use the same params for each elements in figure 8?

**Limitations:**

No. While their should not have any societal impact, further discussions on limitations would be nice.

**Strengths And Weaknesses:**

# strengths
- basically, authors propose to create a suitabl benchmark from multiple dataset directed by subjects and distributions. This approach is interesting, mostly due to the whole framework proposed in this paper.
- the usage of Stochastic weight averaing is really interesting and important.
- nice ablation study
- very nice structure of the paper, especially in section 4


# weaknesses
- authors did not highlight the number of different emotion labels there is for each dataset and, more importantly, for the last model. This only appears in appendix 6 and should be in the core paper.
- some terms are not defined. For instance: eq12, $t$ is undefined ; eq7, $r$ is undefined.
- comparison in section 3.4 is limited to approaches from almost 10 years ago (2017). This would be interesting to compare to more recent ones.
- results are evaluated using accuracy and F1 score. However, emotion recognition is best evaluated using macro F1 especially due to these distributions issues that the authors hoghlights. I think this metric must be added (there is space in table 1).

---

> ### Author Rebuttal · Authors · 2026-03-30
>
> # Response to Reviewer MiK3
> We sincerely thank you for the valuable comments and constructive suggestions on our manuscript. We respond to each concern in detail below and clarify the corresponding revisions we have made.
>
> ---
> >`W1`: The number of emotion labels for each dataset should be included in the main text.
>
> Thank you for this suggestion. The detailed dataset information was originally placed in the appendix due to space limitations. In the revised version, we will include the key dataset statistics in the main text to make the experimental setup clearer and more self-contained.
>
> >`W2`: Some terms are not defined (e.g., $r$ in Eq. (12) and $t$ in Eq. (7)).
>
> We apologize for the lack of clarity. Specifically, $r$ in Eq. (12) denotes the **representation**, and $t$ in Eq. (7) denotes the **current iteration step**. In the revised version, we will systematically define all key notations upon their first occurrence to improve readability.
>
> >`W3`: The comparisons seem limited to older methods (around 2017). Including more recent methods would be more meaningful.
>
> Thank you for this suggestion. We would like to clarify that the baselines are not limited to earlier methods. Specifically, DMMR and MoGE were published in **2024**, while TimeDP, EEGMatch, and EmT were published in **2025**, representing recent advances in the field. We acknowledge that some included baselines (**Mixup** and **MAML**) may still have limitations in terms of recency. However, Mixup and MAML are included due to their methodological relevance as representative **instance-level augmentation** and **meta-learning** approaches, which help better contextualize the performance of MixEmo.
>
> >`W4`: Emotion recognition should preferably be evaluated using macro F1-score.
>
> Thank you for this suggestion. We would like to clarify that our evaluation metrics include Accuracy, Precision, Recall, and F1-score, where the **reported F1-score is indeed the macro F1-score**. We acknowledge that the original description may be ambiguous, and we will clarify this explicitly in the revised version.
>
> >`Q1`: According to Figure 3, the support set and query set appear to be balanced, which may suppress the insertion of $G_{\mathcal{M}}$.
>
> Thank you for this insightful question. We clarify that in HDCL, the support and query sets are **re-partitioned at each iteration**, following a design inspired by **meta-learning**. The partitioning is stochastic, and theoretically, given sufficient iterations, it **does not compromise** the overall distribution integrity of the support and query sets. We acknowledge that this design may cause confusion. In the revised version, we will provide additional implementation details to clarify this process and better address this concern.
>
> >`Q2`: Can $d$ be derived from factors such as dataset size, number of subjects, or number of modalities?
>
> Thank you for this insightful question. In general, it is difficult to analytically derive the optimal value of $d$ directly from such factors. A common practice is to define a reasonable range based on prior knowledge (e.g., data complexity), and then determine the optimal value via parameter sensitivity analysis. Related experiments are provided in **Section 4.4** and **Table 12 in the appendix**.
>
> >`Q3`: What are the t-SNE settings? Are the same parameters used for all elements in Figure 8?
>
> Thank you for your attention to the reliability of the visualization. All t-SNE visualizations are implemented using the TSNE class from the sklearn library. The parameters are set as follows: **n_components=2**, **perplexity=30.0**, **metric="euclidean"**, and **learning_rate="auto"**, with all other parameters **set to default values**. We ensure that all visualizations in Figure 8 use exactly the **same parameter settings**, thereby guaranteeing consistency and fair comparison across different baselines.

---

> > ### Author Rebuttal · Reviewer_MiK3 · 2026-04-01
> >
> > Thank you for the clarifications on the several ambiguous points and for the correction of my mistake about related work dates. Overall, I don't think this fundamentaly changes my review, hence I will keep my score as weak accept.
> > Best

---

> > > ### Author Response · Authors · 2026-04-03
> > >
> > > Dear Reviewer MiK3,
> > >
> > > We sincerely appreciate your insightful review and your dedication to this process. Thanks to your feedback, the quality of our manuscript has been significantly improved.
> > >
> > > Sincerely,
> > > The Authors

---

### Official Review · Reviewer_w1Nv · 2026-03-11

**Soundness:** 2
**Presentation:** 3
**Significance:** 2
**Originality:** 2
**Overall Recommendation:** 4
**Confidence:** 3

**Summary:**

This paper studies cross-subject emotion recognition under heterogeneous data distributions. The authors argue that existing methods are often modality-specific, particularly for EEG, and commonly rely on an i.i.d. assumption that may not hold for real-world emotion data. To address this, they propose MixEmo, which combines Unseen Distribution Generation (UDG) for constructing unseen distribution subsets with Heterogeneous Distribution Collaborative Learning (HDCL) for jointly learning from multiple heterogeneous subsets, together with WGU and SWA for more stable optimization and better generalization. Experiments on five datasets across three modalities show improvements over several data augmentation baselines, although the method is not consistently better than specialized EEG-specific approaches.

**Compliance With Llm Reviewing Policy:**

Affirmed.

**Key Questions For Authors:**

1. Why are broader domain generalization baselines not included, especially for the non-EEG datasets?
2. Beyond the t-SNE plots and case study, could the authors share any additional quantitative evidence supporting the presence of heterogeneous sub-distributions in the training data?
3. The statement in Section 5 that MixEmo “outperforms ... EEG-specific methods” seems too strong, since the reported results do not show consistent improvements over all EEG-specific methods.
4. MixEmo performs augmentation in a frozen representation space. Are the baseline augmentation methods applied in the raw input space or in the same representation space? If not, how is comparison fairness ensured?
5. In Eq. (10), subsets with smaller query losses receive larger weights. Could the authors explain why this design is preferred, instead of emphasizing harder or underrepresented subsets?

**Limitations:**

The proposed method depends on a relatively strong assumption about the existence of clusterable, approximately Gaussian heterogeneous subsets in the representation space. Moreover, the evidence supporting this assumption is mainly qualitative, relying on visualization and case-study analysis rather than stronger quantitative validation. Finally, the empirical evaluation, while fairly broad, does not yet provide enough statistical and comparative evidence to support stronger claims, since variance/significance reporting is limited and the baselines do not include a sufficiently broad set of general domain generalization or robust learning methods.

**Strengths And Weaknesses:**

strength
1. The proposed framework is comparatively extensible across emotion modalities because its key operations are defined at the distribution level in the learned representation space, reducing dependence on EEG-specific designs.
2. The method is conceptually well organized: UDG enhances distributional diversity, while HDCL enables collaborative learning across heterogeneous subsets, with the two components serving complementary roles.
3. The empirical evaluation is fairly broad, covering five datasets across three modalities, together with ablation, sensitivity, convergence, and case-study analyses.

weakness
1. The empirical comparison is not sufficiently broad for a cross-subject generalization paper, as it focuses mainly on data augmentation baselines and a small number of EEG-specific methods, without systematic evaluation against broader domain generalization or robust learning approaches.
2. The evidence for heterogeneous distributions is largely qualitative, relying on t-SNE visualizations and case studies, which is not sufficient to fully justify the GMM-based modeling choice.
3. The theoretical analysis is developed under simplified assumptions, including approximately Gaussian subset distributions, unseen distributions formed by random combinations of known subsets, and convergence under standard smoothness and bounded-variance optimization conditions. These assumptions help motivate the method, but they remain some distance from the non-Gaussian, high-dimensional, and potentially unstable clustering scenarios encountered in practice. As a result, the theory is more supportive than fully explanatory of the empirical behavior of the method.

---

> ### Author Rebuttal · Authors · 2026-03-30
>
> # Response to Reviewer w1Nv
> We sincerely thank you for the valuable comments and constructive suggestions on our manuscript. We respond to each concern in detail below and clarify the corresponding revisions we have made.
>
> ---
> >`W1 & Q1`: Why are broader domain generalization baselines not included, especially for non-EEG datasets?
>
> Based on the technical design of MixEmo, we primarily **position it as a data augmentation** method. Accordingly, we include several representative **augmentation-based** baselines. In addition, to better contextualize the performance of MixEmo, we also incorporate **domain generalization** approaches such as TimeDP, EEGMatch, and MoGE, where TimeDP is a **general-purpose domain generalization** method. Furthermore, MAML as a representative **meta-learning** method.
>
> >`W2 & Q2`: Beyond t-SNE visualizations and case studies, can the authors provide more quantitative evidence to support the existence of heterogeneous sub-distributions in the training data?
>
> We agree that the current evidence can be further strengthened. Therefore, when determining the optimal number of clusters $M$ (**Algorithm 2**), we adopt the **Bayesian Information Criterion (BIC)** as a quantitative metric, where lower values indicate better clustering quality. The BIC values for **different cluster numbers** across datasets are as follows:
>
> |Dataset|3/5|4/6|5/7|6/9|7/9|
> |:---:|---:|---:|---:|---:|---:|
> DEAP|46468149|31313054|**27439410**|29920562|31470551
> SEED|140224237|128528553|**127358620**|128608678|130992876
> WESAD|-178742965|-185499801|**-191452187**|-172232653|-169976806
> SE-R1510|134213859|130323608|**129800922**|138940729|141038570|
> SE-F1530|132108731|130540613|**119436220**|120222853|123705694|
>
> The results indicate that the optimal clustering performance is achieved when $M=5$ or $M=7$ (with WESAD favoring $M=7$), which supports the **existence of heterogeneous subsets** in the data. The t-SNE visualization in Figure 2 is provided as **intuitive support**. Following your suggestion, we will further include BIC trends over varying cluster numbers in the revised version to strengthen the evidence.
>
> >`W3`: The theoretical analysis is based on simplified assumptions and may not fully reflect practical scenarios involving non-Gaussian, high-dimensional, and potentially unstable clustering.
>
> We fully agree with this observation. The gap between simplified assumptions and real-world scenarios may limit the performance of MixEmo. In **Section 3.4 (Comparison and Discussion)**, we have already discussed that "*when feature representations are highly compact within classes and well-separated across classes, the proposed method may have limited ability to generate effective complementary distributions...*". We will incorporate your concern to further clarify the scope and applicability of our method in the revised version.
>
> >`Q3`: The claim in Section 5 that MixEmo “outperforms EEG-specific methods” appears overly strong.
>
> Thank you for pointing out this imprecise wording. We agree that MixEmo does not consistently outperform all EEG-specific methods. In fact, we have already provided a more cautious statement in **Section 4.2.2**: "*Although MixEmo does not comprehensively surpass methods specifically designed for cross-subject EEG emotion recognition…*". We will revise this part in the updated version to ensure that the conclusions are stated more precisely.
>
> >`Q4`: MixEmo performs data augmentation in a frozen representation space. Are baselines applied in the raw input space or the same representation space? How is fairness ensured?
>
> In our setup, for comparisons with data augmentation baselines, all methods are applied in the representation space, i.e., using encoded **sample representations** as input. This ensures that the comparisons are conducted under consistent conditions. We will further provide detailed descriptions of the baselines and experimental configurations in the appendix to improve clarity and transparency.
>
> >`Q5`: In Eq. (10), subsets with smaller query losses are assigned larger weights. Why is this design chosen?
>
> Thank you for pointing out this critical issue. We would like to clarify that there is a mistake in the current formulation of Eq. (10). Our intended design is that subsets with **larger query losses should be assigned higher weights**, in order to accelerate training convergence. Therefore, the loss term **should not include "$-$"**. We have carefully verified the implementation and confirm that all experiments were conducted using this **correct formulation**. Hence, this issue arises from a writing error. We will correct this equation in the revised version. We sincerely appreciate your careful and professional review.

---

> > ### Author Rebuttal · Reviewer_w1Nv · 2026-04-03
> >
> > The response has addressed my previous comments.

---

> > > ### Author Response · Authors · 2026-04-03
> > >
> > > Dear Reviewer w1Nv,
> > >
> > > We sincerely thank you for your valuable comments and constructive suggestions. Your careful and professional review has been highly beneficial in improving the quality of our work, and we greatly appreciate your efforts.
> > >
> > > We are pleased that our responses have adequately addressed your concerns. If you find these revisions convincing and would consider updating your score accordingly, we would be most grateful.
> > >
> > > Thank you again for your support and assistance.
> > >
> > > Sincerely,
> > > The Authors

---

### Official Review · Reviewer_sqqd · 2026-03-13

**Soundness:** 2
**Presentation:** 3
**Significance:** 3
**Originality:** 2
**Overall Recommendation:** 4
**Confidence:** 3

**Summary:**

This paper studies cross-subject emotion recognition and argues that prior methods suffer from two main issues: they are often heavily tailored to EEG-specific designs, and they usually assume i.i.d. data, which may be inappropriate when emotion datasets contain heterogeneous sub-distributions. To address this, the paper proposes MixEmo, a framework with two main components: Unseen Distribution Generation (UDG), which clusters learned representations into multiple distribution prototypes and synthesizes new heterogeneous subsets by combining samples across prototypes; and Heterogeneous Distribution Collaborative Learning (HDCL), which performs collaborative optimization across these subsets using proxy models, weighted query-loss updates, and SWA. The method is evaluated on five datasets spanning EEG, physiological signals, and societal behavior time series, where it outperforms several generic augmentation baselines and is competitive with or slightly better than EEG-specific cross-subject baselines on EEG datasets. The paper also provides ablations, sensitivity analysis, convergence analysis, and a qualitative case study in representation space.

**Compliance With Llm Reviewing Policy:**

Affirmed.

**Final Justification:**

The rebuttal has addressed my concerns. My final justification prefers to accept the paper.

**Key Questions For Authors:**

1. The paper’s central assumption is that the learned representation space contains meaningful heterogeneous sub-distributions. This is mainly supported by t-SNE plots and GMM clustering, which I do not find fully sufficient. Could the authors provide more quantitative evidence that these subsets indeed reflect meaningful distributional heterogeneity, rather than artifacts introduced by the encoder or the visualization itself? For instance, it would be helpful to see whether other clustering diagnostics or analyses in the original feature space lead to a similar conclusion.

2. The current comparisons focus mainly on augmentation baselines and EEG-oriented methods. I think the empirical evaluation would be stronger if the paper also included more general domain generalization or robust learning baselines that are not tied to a specific modality. That would make it easier to judge whether the gains really come from the proposed heterogeneous augmentation and collaborative learning design.

3. There is a typo on line 192: “Table 3” should be “Figure 3.”

**Limitations:**

No. The paper does not adequately discuss limitations or possible negative societal impact. The impact statement is too generic and effectively says no specific consequences need to be highlighted, which is not sufficient for a paper on emotion recognition. The authors can discuss at least:
(1) possible demographic, cultural, and subject-specific biases in emotion datasets;
(2) risks of misuse in surveillance, profiling, or high-stakes human assessment.

**Strengths And Weaknesses:**

The potential strength of this paper: the paper addresses a meaningful and practical problem. Cross-subject generalization remains a key challenge in emotion recognition, and the effort to move beyond EEG-specific designs toward a more modality-agnostic framework is well justified.

My main concern lies in the that the justification relies largely on t-SNE visualizations of learned representations, however, t-SNE can produce visually separated clusters even when the underlying structure is ambiguous, and the paper does not provide sufficient quantitative analysis to confirm that these clusters reflect meaningful subject- or distribution-level differences rather than artifacts of the representation or labels.

---

> ### Author Rebuttal · Authors · 2026-03-30
>
> # Response to Reviewer sqqd
> We sincerely thank you for the valuable comments and constructive suggestions on our manuscript. We respond to each concern in detail below and clarify the corresponding revisions we have made.
>
> ---
> >`Q1`: The heterogeneous sub-distribution assumption is mainly supported by t-SNE visualizations and GMM clustering, which may not be sufficiently convincing.
>
> Thank you for pointing out this important issue. We agree that the current evidence can be further strengthened. Therefore, when determining the optimal number of clusters $M$ (**Algorithm 2**), we adopt the **Bayesian Information Criterion (BIC)** as a quantitative metric, where lower values indicate better clustering quality. The BIC values for **different cluster numbers** across datasets are as follows:
>
> |Dataset|3/5|4/6|5/7|6/9|7/9|
> |:---:|---:|---:|---:|---:|---:|
> DEAP|46468149|31313054|**27439410**|29920562|31470551
> SEED|140224237|128528553|**127358620**|128608678|130992876
> WESAD|-178742965|-185499801|**-191452187**|-172232653|-169976806
> SE-R1510|134213859|130323608|**129800922**|138940729|141038570|
> SE-F1530|132108731|130540613|**119436220**|120222853|123705694|
>
> The results indicate that the optimal clustering performance is achieved when $M=5$ or $M=7$ (with WESAD favoring $M=7$), which supports the **existence of heterogeneous subsets** in the data. The t-SNE visualization in Figure 2 is provided as **intuitive support**. Following your suggestion, we will further include BIC trends over varying cluster numbers in the revised version to strengthen the evidence.
>
> >`Q2`: Including more general domain generalization methods or modality-agnostic robust learning baselines would make the empirical evaluation more convincing.
>
> We fully agree that incorporating a broader range of baselines would make the empirical evaluation more comprehensive and convincing. In our current setup, we include several representative methods. Specifically, TimeDP, EEGMatch, and MoGE are adopted as **domain generalization** approaches, where TimeDP is a **general-purpose domain generalization** method. In addition, Mixup, as a classical **instance-level data augmentation** method, and MAML, as a representative **meta-learning** approach, are also evaluated. Furthermore, we consider **GAN-based augmentation** methods and **graph neural network**-based approaches to provide comparisons from diverse technical perspectives. That said, we acknowledge that the current set of baselines can still be further expanded. In future work, we will incorporate additional categories of methods to enable a more comprehensive evaluation and more accurately position the performance of MixEmo.
>
> >`Q3`: There is a typo in line 192: “Table 3” should be “Figure 3”.
>
> Thank you for your careful review. We have carefully checked the manuscript and confirm that the correct reference is **“Table 3” rather than “Figure 3”**. Table 3 presents the detailed configuration of MixEmo (see **Appendix C**). We kindly ask whether this discrepancy might be due to a version difference. We will also further verify all cross-references in the revised version to ensure consistency.
>
> >`Q4`: The paper does not sufficiently discuss its limitations or potential negative societal impacts.
>
> We fully agree that a thorough discussion of methodological limitations and potential societal impacts is essential.
>
> 1. Methodological limitations. In **Section 3.4 (Comparison and Discussion)**, we have discussed that "*when feature representations are highly compact within classes and well-separated across classes, the proposed method may have limited capacity to generate effective complementary distributions...*". To improve clarity, we will introduce a dedicated “Limitations” section in the revised version to present these aspects more systematically.
> 2. Potential societal impacts. We acknowledge that the current **Impact Statement** lacks specificity. In the revised version, we will substantially expand this part, focusing on:
>
>     * **Data bias and generalization risks**. Emotion datasets may contain biases related to demographics, cultural backgrounds, or specific populations, which may limit generalization and lead to degraded performance or systematic bias for underrepresented groups.
>     * **Potential misuse in sensitive scenarios**. Emotion recognition technologies may be applied in high-risk contexts such as surveillance, profiling, or automated decision-making. Without proper ethical oversight, transparency, and informed consent, such applications may raise privacy concerns and fairness issues. We will explicitly state that this work should not be deployed in high-risk or sensitive scenarios without strict ethical and regulatory safeguards.

---

> > ### Author Rebuttal · Reviewer_sqqd · 2026-04-04
> >
> > Thanks for your effort. My concerns have been adequately addressed.

---

> > > ### Author Response · Authors · 2026-04-04
> > >
> > > Dear Reviewer sqqd,
> > >
> > > We sincerely thank you for your valuable comments and constructive suggestions. We are pleased that our responses have adequately addressed your concerns.
> > >
> > > If you find these responses convincing and would consider revising your score accordingly, we would be most grateful.
> > >
> > > Sincerely,
> > > The Authors

---

### Official Review · Reviewer_cQUb · 2026-03-16

**Soundness:** 2
**Presentation:** 1
**Significance:** 3
**Originality:** 2
**Overall Recommendation:** 3
**Confidence:** 4

**Summary:**

The paper proposes a modality-agnostic model to handle data with heterogeneous distributions. The approach introduces two main blocks: Unseen Distribution Generation and Heterogeneous Distribution Collaborative Learning (HDCL).

**Compliance With Llm Reviewing Policy:**

Affirmed.

**Final Justification:**

I question the generalization of the approach.

**Key Questions For Authors:**

Why does the evaluation not consider cross-corpus scenarios?

Can you confirm that you are training and testing the models separately for each database?

**Strengths And Weaknesses:**

Strenghs

The results are encouraging. The approach addresses a machine learning problem that has applications to many other problems.

Major limitations

First, I thought the same model was used across all modalities, training the framework on all the databases. If this were the case, I would question the principle. Each modality has its own characteristics, so building a modality-agnostic approach is questionable. There is no one-size-fits-all strategy, especially for emotion recognition. The algorithm architecture that works for speech processing should be quite different from the EEG data. It is not the distribution, but the temporal resolution, the spectral representation, etc.

Based on the experiments, it seems that each database is trained independently. If this is the case, I just don’t understand the problem. Why should we even care that modalities are different if we are going to use in-domain data for training?


This interpretation in Figure 2 is questionable. First of all, you are projecting a high-dimensional space into a 2D space. The data can be perfectly clustered in a higher-dimensional space even when the 2D representation does not indicate it. If you are comparing figures across databases, this analysis is also not meaningful, given that diﬀerent mappings are created for each database. Therefore, the comparisons are not apples to apples.

The first and second order statistics do not determine if the combination of databases creates a heterogeneous space. Notice that Gaussian distributions are parametrized by mean and standard deviation. This is not the case for other distributions, so the formulation is not general

No cross-corpus evaluation. The testing sets are much easier than scenarios where subjects are recorded under different conditions. This latter scenario is the most interesting for evaluating model generalization.


Minor limitations


“Therefore, the UDG can effectively generate representation sets that follow unseen distributions, thereby improving overall distributional diversity of the data.” How do you conclude this?

---

> ### Author Rebuttal · Authors · 2026-03-30
>
> # Response to Reviewer cQUb
> We sincerely thank you for the valuable comments and constructive suggestions on our manuscript. We respond to each concern in detail below and clarify the corresponding revisions we have made to address these issues.
>
> ---
> >`Q1`: Are the models trained and tested separately on each dataset? If the training is conducted within the same domain, why is it still necessary to consider modality differences?
>
> 1.**Training setup**. All experiments are conducted **within each dataset**, i.e., training and testing are performed independently without cross-dataset training.
> 2.**Necessity of modality-weak dependence**. Multimodal emotion recognition has become a key research direction, where a central step is the **alignment** and **fusion** of representations across modalities. A typical pipeline can be summarized as:
> ```
> Emotion modality (modality-specific augment, EXISTING) → Modality-specific encoder → Alignment&Fusion (representation augment, OURS) → Downstream task
> ```
> In such a setting, **multiple modality-specific augmentation strategies are required**, which may lead to:
>   * Increased training instability;
>   * Higher hyperparameter tuning complexity.
>
> To address these issues, we focus on **representation-level augmentation** and propose a relatively general mechanism to complement modality-specific approaches, thereby improving applicability and scalability in multimodal scenarios.
>
> >`Q2`: The interpretation in Figure 2 is debatable, as data may exhibit well-separated clusters in high-dimensional space.
>
> We agree that clustering structures observed via t-SNE may not faithfully reflect the true distribution in high-dimensional space. Therefore, when determining the optimal number of clusters $M$ (**Algorithm 2**), we adopt the **Bayesian Information Criterion (BIC)** as a quantitative metric, where lower values indicate better clustering quality. The BIC values for **different cluster numbers** across datasets are as follows:
>
> |Dataset|3/5|4/6|5/7|6/9|7/9|
> |:---:|---:|---:|---:|---:|---:|
> DEAP|46468149|31313054|**27439410**|29920562|31470551
> SEED|140224237|128528553|**127358620**|128608678|130992876
> WESAD|-178742965|-185499801|**-191452187**|-172232653|-169976806
> SE-R1510|134213859|130323608|**129800922**|138940729|141038570|
> SE-F1530|132108731|130540613|**119436220**|120222853|123705694|
>
> The results indicate that the optimal clustering performance is achieved when $M=5$ or $M=7$ (with WESAD favoring $M=7$), which supports the **existence of heterogeneous subsets** in the data. The t-SNE visualization in Figure 2 is provided as **intuitive support**. Following your suggestion, we will further include BIC trends over varying cluster numbers in the revised version to strengthen the evidence.
>
> >`Q3`: Why is cross-corpus evaluation not considered?
>
> We agree that cross-corpus evaluation provides a more stringent test of generalization. This work **focuses on cross-subject emotion recognition**, and our experimental setup and dataset selection **follow the standard paradigm**. Specifically, DEAP and SEED are widely used EEG datasets for cross-subject evaluation; WESAD is a physiological dataset covering multiple experimental scenarios (e.g., speaking, reading, and watching videos); and SE-R1510 and SE-F1530 are societal behavior datasets that include diverse patterns such as solitary events, social interactions, and research behaviors. Importantly, in WESAD, SE-R1510, and SE-F1530, individual subjects typically cover only a **subset of scenarios**, and there exist **significant differences in behavioral patterns**. As a result, under cross-subject splits, the training and test sets naturally form an **implicit cross-corpus setting**. For example, different subjects may reside in distinct living environments or exhibit different behavioral tendencies (e.g., socially active vs. more solitary). Therefore, there exists a **systematic distribution shift** between the training and test sets.
>
> >`Q4`: How do you justify the claim that “UDG can generate representations following unseen distributions, thereby improving overall distribution diversity”?
>
> UDG first employs a Gaussian Mixture Model to separate heterogeneous subsets and characterizes each subset using its mean and variance, which are treated as distribution prototypes. Based on these prototypes, UDG constructs unseen distributions by randomly combining different prototypes. Specifically, by **adjusting the ratio between a primary subset and auxiliary subsets**, the resulting distributions have means and variances that **do not coincide with any single prototype**. This mechanism enables UDG to generate representations following unseen distributions, thereby enhancing overall distribution diversity. This conclusion is formally stated in **Proposition 2.1**, with the proof provided in **Appendix A.1**. We will further clarify this part in the main text to improve readability.

---

> > ### Author Rebuttal · Reviewer_cQUb · 2026-04-03
> >
> > I question the generalization of the approach.

---

> > > ### Author Response · Authors · 2026-04-04
> > >
> > > We are pleased that our previous response has addressed your concerns, and we sincerely thank you for your further valuable comments regarding the generalizability. In response to this concern, we provide additional clarification from both the perspectives of **technical principles** and **empirical validation**, explaining why MixEmo demonstrates stronger generalizability than modality-specific cross-subject emotion recognition methods.
> > >
> > > 1. **Technical principles**. MixEmo enhances model generalization through **Unseen Distribution Generation (UDG)** and **Heterogeneous Distribution Collaborative Learning (HDCL)**. Both key components operate at the **representation level** rather than being designed for specific modalities. Therefore, the proposed method can be flexibly integrated with different types of encoders, making it applicable across multiple emotion modalities.
> > >
> > > 2. **Empirical validation**. To verify the effectiveness of MixEmo across different modalities, we conduct experiments on five datasets covering three representative emotion modalities: **EEG (3D tensors)**, **physiological signals (high-dimensional vectors)**, and **societal behavior (multivariate time series)**. The results show that MixEmo consistently outperforms both general-purpose data augmentation methods and modality-specific cross-subject emotion recognition approaches on all datasets.
> > >
> > > In summary, as MixEmo performs data augmentation at the **representation level**, it exhibits **weak modality dependence** and can serve as a **plug-and-play** strategy to improve the performance of various backbones in cross-subject emotion recognition tasks.
> > >
> > > We hope the above clarifications adequately address your concerns regarding generalizability. If you find these explanations convincing and would consider revising your score accordingly, we would be most grateful. We would also be happy to further discuss any additional questions you may have.

---

### Decision · Program_Chairs · 2026-04-30

**Decision:**

Accept (regular)

**Comment:**

This paper studies cross-subject emotion recognition under heterogeneous data distributions and proposes MixEmo by combining Unseen Distribution Generation (UDG) for constructing unseen distribution subsets with Heterogeneous Distribution Collaborative Learning (HDCL) for jointly learning from multiple heterogeneous subsets. Four reviewers reviewed this paper with the overall ratings of three weak accept and one weak reject. The reviewers recognized the meaningful and practical problem, good organization and presentation, interesting method, and nice ablation studies. Meanwhile, they also pointed out several critical concerns about different aspects of this paper, such as no cross-corpus evaluation, questionable figures, outdated baselines, insufficient evaluation metrics, insufficient empirical comparison, insufficient quantitative analysis, and too strong theoretical assumptions. The reviewers provided rebuttal to answer the reviewers' questions. After the rebuttal, three reviewers are generally satisfied with the revision, while one reviewer still has concerns on the generalization of the approach. The authors further explained the generalization from both the perspectives of technical principles and empirical validation, which the AC believes is convincing. Therefore, the AC recommends weak acceptance and encourage the authors to revise this paper accordingly.